# GeoPT: Scaling Physics Simulation via Lifted Geometric Pre-Training

Haixu Wu [* 1]  Minghao Guo [* 1]  Zongyi Li [1]  Zhiyang Dou [1]  Mingsheng Long [2]  Kaiming He [1]  Wojciech Matusik [1]

## Abstract

Neural simulators promise efficient surrogates for physics simulation, but scaling them is bottlenecked by the prohibitive cost of generating high-fidelity training data. Pre-training on abundant off-the-shelf geometries offers a natural alternative, yet faces a fundamental gap: supervision on static geometry alone ignores dynamics and can lead to negative transfer on physics tasks. We present **GeoPT**, a unified pre-trained model for general physics simulation based on *lifted geometric pre-training*. The core idea is to augment geometry with synthetic dynamics, enabling dynamics-aware self-supervision without physics labels. Pre-trained on over one million samples, GeoPT consistently improves industrial-fidelity benchmarks spanning fluid mechanics for cars, aircraft, and ships, and solid mechanics in crash simulation, reducing labeled data requirements by 20-60% and accelerating convergence by 2×. These results show that lifting with synthetic dynamics bridges the geometry-physics gap, unlocking a scalable path for neural simulation and potentially beyond. Code is available at https://github.com/Physics-Scaling/GeoPT.

## 1. Introduction

Neural simulators have emerged as efficient surrogates for classical numerical solvers, accelerating physics simulation across scientific discovery and engineering design (Li et al., 2021; Wang et al., 2023; Zhou et al., 2024). By learning an operator that maps geometry and initial conditions to solution fields, these models reduce complex evaluations to a single forward pass, substantially lowering inference costs. This amortization is particularly vital for iterative design systems, as evidenced by the widespread adoption of commercial neural simulation software within industrial

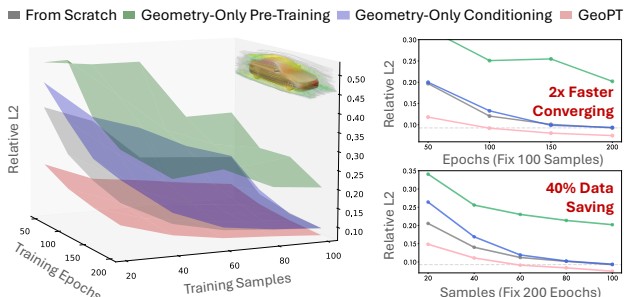

*Figure 1.* Neural aerodynamics simulation on DrivAerML (Ashton et al., 2024) based on Transolver (Wu et al., 2024) backbone. Geometry-only pre-training and conditioning refer to pre-training by predicting vector distance (Faugeras & Gomes, 2000) of given positions and utilizing geometry representation extracted by Hunyuan3D (Tencent, 2025) as auxiliary feature, respectively.

design (Ansys Inc., 2026; Altair Engineering Inc., 2026a).

Achieving industrial-fidelity accuracy, however, typically hinges on scaling model capacity and training data. Unlike vision or language, where well-established self-supervision learning methods and web-scale data enable such scaling (He et al., 2022; Achiam et al., 2023), neural simulators remain dominated by *supervised learning* on physics data generated from numerical solvers. The primary scaling bottleneck is label generation: each training sample requires a full numerical solve, whose computational cost increases sharply with geometry and physics complexity. For example, in the DrivAerML aerodynamics dataset (Ashton et al., 2024), generating a single industrial-fidelity sample can cost $6.1 \times 10^4$ CPU-hours. This prohibitive cost severely limits the scaling of neural simulators across diverse physics.

In this paper, we explore *self-supervised pre-training* for neural simulation as a pathway to scale beyond solver-simulated datasets. Fundamentally, the solution field of a physical system is jointly determined by geometry and dynamics: the geometry defines the spatial domain and boundaries, while dynamics specify how the system is driven. Although physics labels from geometry-dynamics coupled simulations are costly to obtain, the geometry alone is abundantly available at web scale from public repositories (Chang et al., 2015; Deitke et al., 2023). This asymmetry motivates a *geometric pre-training* paradigm: we pre-train neural simulators only on geometry data and introduce simulated physics labels during downstream fine-tuning.

*Equal contribution  [1]MIT CSAIL  [2]Tsinghua University. <wuhaixu98@gmail.com>, <guomh2014@gmail.com>.

*Proceedings of the 43rd International Conference on Machine Learning*, Seoul, South Korea. PMLR 306, 2026. Copyright 2026 by the author(s).

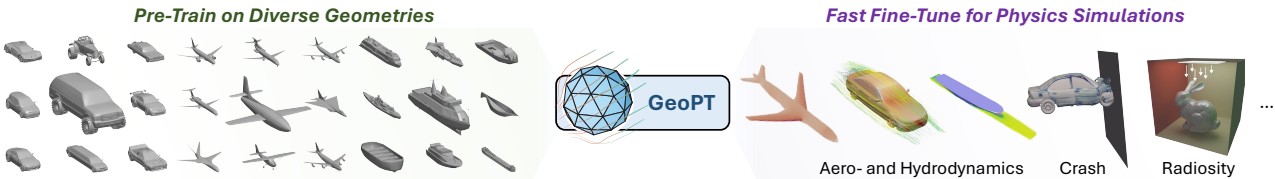

*Figure 2.* GeoPT offers a way to scale up neural simulators with off-the-shelf geometries and enables fast fine-tuning for various physics.

The central challenge is that previous self-supervised learning methods only optimize the model within the *native space* of the pre-training data, such as learning to reconstruct from masked images (He et al., 2022) or learning to identify similar samples from augmentations (Chen et al., 2020), where the model input will not exceed the original information of pre-training data. This paradigm succeeds when the pre-training data space is aligned with downstream tasks, such as image pre-training for recognition. However, a fundamental gap emerges when the downstream task inhabits a space strictly richer than that of the pre-training data. Neural simulation exemplifies this failure: while pre-training on geometry data is viable, the downstream task requires representations that encode the coupled interaction of geometry and dynamics, while geometry-only pre-training can only learn a reduced representation. As evidenced empirically in Fig. 1, geometry-only supervision for pre-training cannot benefit or even degrade downstream performance.

To bridge this gap, we propose *dynamics-lifted geometric pre-training*, defining supervision within an expanded space that reflects the geometry-dynamics coupling required by downstream tasks. Specifically, we augment the geometry-only pre-training input with randomly sampled velocity fields as dynamics conditions and leverage the dynamics-induced, geometry-bounded transport trajectories as self-supervision. Such pre-training can go beyond the original geometry space and enables the model to learn geometry-dynamics coupled representations in a lifted space. After pre-training, the model is fine-tuned to specific physics tasks by specializing the dynamics condition with the corresponding simulation settings and learning from solver-generated labels. In effect, this *lifted self-supervision* learns a dynamics-aware prior from massive unlabeled geometry, scaling without task-specific simulation during pre-training.

We perform dynamics-lifted geometric pre-training on a large-scale 3D geometry repository (Chang et al., 2015), generating over one million solver-free geometric-walk samples by sampling dynamics conditions for each shape. We refer to the resulting geometric pre-trained model as **GeoPT**. GeoPT consistently improves downstream accuracy and data-efficiency across industrial-fidelity fluid and solid benchmarks, including car, aircraft aerodynamics, ship hydrodynamics and crash simulation (Figs. 1-2), which reduces physics data requirements by 20-60%, accelerates convergence by up to $2\times$ and can generalize to boarder

physics domains, e.g., radiosity, successfully unlocking the scaling benefits of neural simulators.

## 2. Related Work

### 2.1. Neural Simulators

Our review focuses on the evolution of neural simulators, which we categorize into model architectures and foundation models. For classical numerical solvers, we refer readers to established reviews (Solin, 2005; Jasak, 2009).

**Model architectures.** Extensive architectures have been explored for neural simulation (Raissi et al., 2020; Pfaff et al., 2021; Li et al., 2025), with neural operators representing notable progress by formalizing simulation as learning maps between function spaces (Kovachki et al., 2023) to solve PDEs. FNO (Li et al., 2021) and its variants (Wen et al., 2022; Li et al., 2023a; Rahman et al., 2023; Wu et al., 2023) approximate integral operator via linear transformations in Fourier space. Transformers (Vaswani et al., 2017) have also been adopted (Li et al., 2023c; Wu et al., 2024), where the attention mechanism serves as a global integral operator (Kovachki et al., 2023), and naturally accommodates irregular geometries by treating mesh points as tokens. To address the quadratic complexity in geometric resolution, efficient attention (Choromanski et al., 2021) and its variants are introduced to Transformer-based simulators (Cao, 2021; Hao et al., 2023). Recently, Transolver (Wu et al., 2024) bypasses mesh structure by learning latent physical states, demonstrating strong performance and scaling in industrial design applications (Luo et al., 2025; Nabian et al., 2025). In this paper, we use Transolver as our backbone, although the proposed pre-training method is architecture-agnostic.

**Physics foundation models.** Scaling neural simulators as physics foundation models has been explored to improve simulation performance (Yang et al., 2023; Herde et al., 2024; Ye et al., 2024). Poseidon pre-trains on 2D fluid dynamics with temporal conditioning (Herde et al., 2024); DPOT expands to diverse physics with auto-regressive prediction (Hao et al., 2024); Unisolver incorporates PDE information via conditional architectures (Zhou et al., 2025); and P3D extends to 3D fluids (Holzschuh et al., 2026). Despite these advances, existing models remain restricted to a specific physics family on regular grids and do not generalize to

the industrial simulations evaluated in this paper. Moreover, they still rely on compute-heavy simulation data for scaling, whereas GeoPT pre-trains on off-the-shelf geometries alone, offering a more scalable path to large-scale pre-training.

## 2.2. Self-Supervised Pre-Training

Self-supervised pre-training has achieved remarkable success in vision (Caron et al., 2021; He et al., 2022; Lin et al., 2023; Tencent, 2025), language (Devlin et al., 2019; Achiam et al., 2023), and audio (Huang et al., 2022), typically by constructing pretext tasks from raw input to learn transferable representations that mitigate labeling bottlenecks.

Well-established methods include reconstructing masked inputs (Vincent et al., 2008; He et al., 2022; Xie et al., 2022) and predicting similarity among augmentations (He et al., 2020; Chen et al., 2020; Caron et al., 2021). Despite the diverse pretext tasks, these methods limit the learning process within the pre-training data space without introducing external information. This paradigm is widely adopted for input-aligned pre-training and fine-tuning tasks. Our setting differs fundamentally: we pre-train on geometry yet target generalization to the higher-dimensional physics space, where native-space pre-training may collapse due to potential randomness of uncovered factor, e.g., dynamics.

Self-supervised pre-training has also been explored in 3D geometry understanding (Yu et al., 2022; Tencent, 2025), and recent work incorporates such pre-trained encoders as auxiliary feature extractors for physics-learning (Deng et al., 2024; Zhang et al., 2026). However, these approaches rely on frozen geometric encoders and do not scale the core physics backbone. Furthermore, since pre-training uses static geometric supervision only, the learned representations lack awareness of dynamics. In contrast, we directly pre-train the physics backbone itself and bridge the geometry-physics gap through dynamics-lifted supervision.

## 3. Problem Setup

We consider physics systems defined with geometric objects $G \in \mathcal{G}$, where $\mathcal{G}$ denotes the space of geometries in $\mathbb{R}^C$, and system conditions $S \in \mathcal{S}$ that specify how the system is driven, including boundary types, external forces, governing equation, initial states, etc. The numerical simulator produces a solution $\boldsymbol{u}: \mathbb{R}^C \to \mathbb{R}^{C_{\mathbf{u}}}$, where $\boldsymbol{u}(\mathbf{x})$ is the corresponding physics quantities on discretized mesh points $\mathbf{x} \in \mathbb{R}^C$ and $C_{\mathbf{u}}$ denotes the number of physical variables. In this work, we focus on steady-state simulation, a primary paradigm for industrial design and large-scale engineering analysis (Azizzadenesheli et al., 2024). For example, in aerodynamics, $G$ is a car surface mesh, $S$ specifies incoming flow velocity and direction, $\mathbf{x}$ denotes the query point, and $\boldsymbol{u}(\mathbf{x})$ contains the resulting pressure and velocity fields.

The neural simulator $\mathcal{F}_\theta(G, S)$ learns to estimate the physical quantities directly, bypassing the expensive numerical solvers. The learning objective is to minimize:

$$\mathcal{L}^{\text{physics}} = \mathbb{E}_{\mathcal{D}} \left[ \|\mathcal{F}_\theta(\mathbf{x}; G, S) - \boldsymbol{u}(\mathbf{x})\|_2^2 \right]. \quad (1)$$

Previous supervised learning relies on labeled dataset $\mathcal{D} = \{(\mathbf{x}, G, S, \boldsymbol{u}(\mathbf{x}))\}$. The geometric pre-training studied in this paper seeks an initialization $\widehat{\theta}$ for $\mathcal{F}_\theta$ from unlabeled geometries $\mathcal{G}$ alone, so that the downstream optimization of Eq. (1) converges faster with fewer physics labels.

## 4. Method

We aim to pre-train a general neural simulator solely from geometric data. This requires a supervision signal that reflects the geometry-dynamics coupling of downstream tasks, while enabling adaptation to diverse simulations. GeoPT achieves this through lifted geometric pre-training, which naturally yields a unified interface for varied physics tasks.

### 4.1. Lifting Geometry to Physics

**The geometry-physics gap.** Given the abundance of unlabeled 3D shapes, a natural strategy is to pre-train neural simulators using supervision derived purely from geometry. A straightforward approach trains the model to predict geometric features at query points $\mathbf{x}$ with following loss:

$$\mathcal{L}^{\text{pre}}_{\text{native}} = \mathbb{E}_{\mathbf{x}, G} \left[ \|\mathcal{F}_{\widehat{\theta}}(\mathbf{x}; G) - \boldsymbol{h}_G(\mathbf{x})\|_2^2 \right], \quad (2)$$

where $\boldsymbol{h}_G(\mathbf{x}) \in \mathcal{H}$ denotes the self-supervision target at $\mathbf{x}$, with $\mathcal{H}$ being the space of geometric features such as occupancy, signed distance (SDF), or vector distance fields (Faugeras & Gomes, 2000). We refer to this as *native pre-training*: the model learns a mapping $\mathcal{G} \to \mathcal{H}$, where the supervision $\boldsymbol{h}_G(\mathbf{x})$ is derived solely from the geometry, determined by the static spatial information $G$.

Despite the intuition that pre-training should help, the objective in Eq. (2) does not reliably benefit neural simulation in practice. We empirically study this on the aerodynamics task in DrivAerML (Ashton et al., 2024) using Transolver (Wu et al., 2024). Quantitatively, as shown in Fig. 1, native pre-training followed by fine-tuning on physics labels degrades accuracy compared to training from scratch by a large margin. To understand this failure, we visualize the learned aggregation weights across spatial points in Transolver[1], which represent the spatial correlations the model has learned and can reflect the potential interactions among different positions under the physical simulation context. We compare: (i) training with physics supervision, and (ii) training with geometry-only supervision from Eq. (2). As

---

[1]Transolver aggregates mesh points into several internally representation-consistent tokens. If two positions are more likely to be ascribed to the same token, they are learned to be correlated.

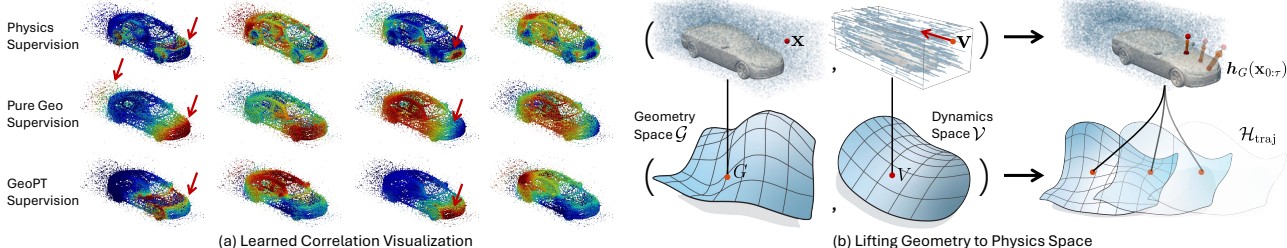

*Figure 3.* Geometry-physics analysis. (a) Visualization of learned correlations on DrivAerML (Ashton et al., 2024). We train Transolver (Wu et al., 2024) using different supervisions and visualize the spatial distribution of learned aggregation weights in four tokens. Brighter colors indicate higher token assignment likelihood, revealing correlations captured by the model. See Appendix F full results. (b) We lift the geometry space by augmenting it with synthetic velocity fields, which further derive a dynamics-aware supervision.

shown in Fig. 3(a), the two settings yield starkly different patterns. With geometry-only supervision, the model groups regions by static shape cues, assigning both front and back volumes to the same state and producing left-right asymmetric patterns. In contrast, physics supervision yields front-back asymmetric and left-right symmetric patterns that align with the aerodynamic flow structure.

The root cause of this failure becomes clear when comparing native pre-training (Eq. (2)) with the downstream task (Eq. (1)). Downstream prediction depends jointly on geometry $G$ and dynamics conditions $S$, yet native pre-training involves only $G$, where dynamics are entirely absent. Without any notion of dynamics, native pre-training cannot capture the geometry-dynamics coupling in physics simulation.

**Parameterize dynamics.** To bridge this gap, we need to design a pre-training objective that incorporates dynamics while relying only on geometry data: the supervision should remain geometry-derived, yet the learning process should also encode dynamics. A key question is how to represent dynamics $S$ in a form amenable to self-supervised learning.

We begin by examining how particles behave in physical systems. In any dynamic process, a particle at position $\mathbf{x}$ is not static but moves under the governing physics. This evolution can be characterized by an (instantaneous) velocity field $\boldsymbol{v}_S : \mathbb{R}^C \times \mathbb{R} \to \mathbb{R}^C$ determined by the simulation settings $S$, which can be formalized as:

$$\frac{\mathrm{d}\mathbf{x}_t}{\mathrm{d}t} = \boldsymbol{v}_S(\mathbf{x}_t, t) \cdot \mathbb{1}_G(\mathbf{x}_t), \quad \mathbf{x}_0 = \mathbf{x}, \qquad (3)$$

where $\mathbb{1}_G(\cdot)$ equals 0 inside or on the boundary $G$ and 1 otherwise. This formulation encodes two key structures of physical simulation: (i) The velocity field $\boldsymbol{v}_S$ couples spatially distant points: trajectories from different initial positions may intersect, causing these points to share correlated physical responses, mirroring how quantities at different locations become coupled through shared flow or force transmission. (ii) The indicator $\mathbb{1}_G$ halts trajectories at the geometry boundary, reflecting that physical responses are fundamentally shaped by boundary interactions, e.g., surface pressure in aerodynamics, contact forces in crash

simulation, or radiosity in light transport. This formulation is generic across physical regimes: in fluid dynamics, $\boldsymbol{v}_S$ relates to the flow velocity; in solid mechanics, it describes the displacement; in radiative transport, it represents propagation direction. The velocity field $\boldsymbol{v}_S$ thus provides a fundamental parameterization of dynamics conditions $S$.

**Lifting geometry via synthetic dynamics.** This observation offers a path to incorporating dynamics into pre-training. Rather than relying on the physics-determined $\boldsymbol{v}_S$, which requires expensive simulation to obtain, we construct *synthetic velocities* by randomly sampling per-particle velocity:

$$\frac{\mathrm{d}\mathbf{x}_t}{\mathrm{d}t} = \mathbf{v} \cdot \mathbb{1}_G(\mathbf{x}_t), \quad \mathbf{x}_0 = \mathbf{x}, \quad \mathbf{v} \sim \mathrm{Unif}(\mathbb{B}^C), \quad (4)$$

where $\mathbb{B}^C = \{\mathbf{v} \in \mathbb{R}^C : \|\mathbf{v}\|_2 \leq v_{\max}\}$. Denote $V \in \mathcal{V}$ as the collection of such per-point velocities across all query points as shown in Fig. 3(b).

The self-supervision target then becomes the trajectory of geometric features under this synthetic dynamics:

$$\boldsymbol{h}_G(\mathbf{x}_{0:\tau}) = \{\boldsymbol{h}_G(\mathbf{x}_t)\}_{t=0}^{\tau} \in \mathcal{H}_{\mathrm{traj}}, \qquad (5)$$

where $\tau$ is a given fixed time horizon. By tracking how geometric features evolve along these synthetic trajectories, we obtain a *dynamics-aware* supervision signal constructed entirely from geometry.

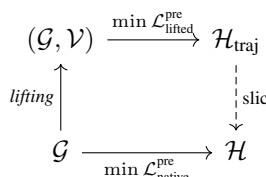

In effect, we *lift* the pre-training from the native geometry space to a joint geometry-dynamics space. The inset diagram illustrates the key relationships: the vertical arrow $\mathcal{G} \to (\mathcal{G}, \mathcal{V})$ is the *lifting* operation, augmenting each geometry with random velocity fields; the top arrow $(\mathcal{G}, \mathcal{V}) \to \mathcal{H}_{\mathrm{traj}}$ is the *lifted pre-training* task, predicting feature trajectories under synthetic dynamics; the bottom arrow $\mathcal{G} \to \mathcal{H}$ is *native pre-training*; and the dashed arrow $\mathcal{H}_{\mathrm{traj}} \to \mathcal{H}$ is *slicing*, which recovers static features by taking $t = 0$. Native pre-training is thus a degenerate case of lifted pre-training: when dynamics are removed, the trajectory collapses to a single point.

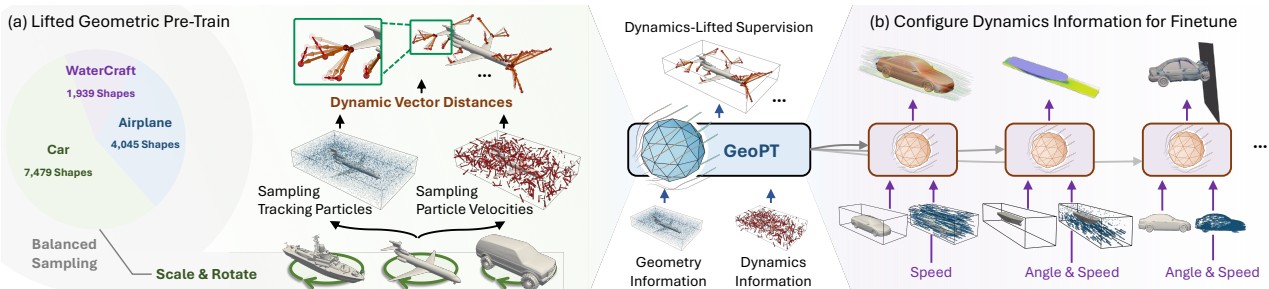

*Figure 4.* Overall design of GeoPT. (a) To ensure the pre-training diversity, we pre-train the model with geometry randomly sampled from the public repository (Chang et al., 2015) and generate the supervision for random tracking points under random dynamics. (b) Through a dynamics-lifted framework, we can configure the dynamics condition to "prompt" the corresponding pre-training capability of GeoPT.

Crucially, downstream simulation also operates on the joint space $(\mathcal{G}, \mathcal{V})$, representations learned via lifting therefore transfer directly to physics simulation tasks.

*Remark* 4.1 (**Theoretical interpretation**). The trajectory evolution in Eq. (4) corresponds to solving a fundamental mass-conservation system, described by the transport equation with sticking boundary: $\partial_t f + v \cdot \nabla_x f = 0$, where $f(x, v, t)$ is a phase-space density. Pre-training with randomly sampled velocity fields $v \in \mathcal{V}$ can be viewed as learning to obey the conservation law under arbitrary dynamics, providing a universal prior for the downstream physical simulation. See Appendix A for details.

### 4.2. Lifted Geometric Pre-Training

The above-described dynamics-lifted framework provides both a pre-training objective and a unified interface for downstream tasks: the model receives geometry and velocity as input during both pre-training and fine-tuning. We now present the complete GeoPT system.

**Pre-training objective.** Following the lifting perspective, we pre-train the model to predict geometric feature trajectories under synthetic dynamics:

$$\mathcal{L}_{\text{lifted}}^{\text{pre}} = \mathbb{E}_{\mathbf{x}, G, V} \left[ \left\| \mathcal{F}_{\widehat{\theta}}(\mathbf{x}; G, V) - \boldsymbol{h}_G(\mathbf{x}_{0:\tau}) \right\|_2^2 \right]. \quad (6)$$

The expectation is over three sources of variation (Fig. 4): (i) geometry $G$ sampled with category-balanced sampling from the geometry dataset $\mathcal{G}$, (ii) tracking point initial position $\mathbf{x}$ sampled from both the surrounding volume space $\Omega_G$ and geometry boundary $G$, and (iii) per-point velocity $\mathbf{v} \in V$ sampled uniformly from a bounded ball $\mathbb{B}^C$. Given geometry-dynamics coupled information $(G, V)$, the trajectory $\mathbf{x}_{0:\tau}$ is deterministically computed via Eq. (4), and the supervision target $\boldsymbol{h}_G(\mathbf{x}_{0:\tau}) = \{\boldsymbol{h}_G(\mathbf{x}_t)\}_{t=0}^{\tau}$ is the sequence of geometric features along this path.

The composition of these three factors yields a *combinatorially* large pre-training space: each geometry admits infinitely many query positions, and each position can be paired with arbitrary velocities, enabling massive data generation from a finite set of shapes. The model $\mathcal{F}_{\widehat{\theta}}$ receives query position $\mathbf{x}$, velocity $V$, and geometry $G$, and outputs a feature trajectory instead of static geometry features. Thus, the whole learning process lies in a geometry-dynamics coupled space, which is higher-dimensional than native space.

As illustrated in Fig. 4, we pre-train on industry-relevant subsets of ShapeNet (Chang et al., 2015) (cars, airplanes, watercraft), comprising over 10,000 unique geometries. Although these shapes differ from industrial models, they provide foundational knowledge of real-world geometry. Coupled with multiple dynamic trajectories per geometry, our pre-training leverages 1,346,300 training samples in total.

**Fast Fine-tuning.** After pre-training, GeoPT captures physics-aligned correlations conditioned on the velocity. As shown in Fig. 4(b), GeoPT adapts to downstream simulation tasks by replacing randomly sampled velocities with task-specific velocity $V_S = \{\mathbf{v}_S\}$ that encode the simulation settings $S$ from Eq. (1). The fine-tuning objective is:

$$\mathcal{L}^{\text{fine}} = \mathbb{E}_{\mathbf{x}, G, V_S} \left[ \left\| \mathcal{F}_\theta(\mathbf{x}; G, V_S) - \boldsymbol{u}(\mathbf{x}) \right\|_2^2 \right]. \quad (7)$$

The key is configuring $V_S$ to explicitly encode simulation settings $S$ for each domain. Here are examples for Table 1:

(i) Aerodynamics: $S$ specifies incoming flow conditions, including angle of attack, sideslip angle, and freestream velocity. We encode these as $V_S$ with direction aligned to the flow and magnitude equal to the freestream speed. This covers both car and airplane simulations.

(ii) Hydrodynamics: $S$ specifies vessel speed and water-air interface conditions. We configure separate $V_S$ for water and air phases, with directions and magnitudes reflecting the two-phase flow in ship resistance simulation.

(iii) Crash simulation: $S$ specifies impact location, direction, and material properties. We encode these as $V_S$ with direction aligned to the impact and spatially decaying magnitude from the collision point, reflecting force propagation.

This unified interface, geometry $G$ plus velocity field $V_S$, allows a single pre-trained model to adapt to diverse physics

by reconfiguring the velocity input.

*Remark* 4.2 (**Native space v.s. lifted pre-training**). By comparing native space (Eq. (2)) and lifted (Eq. (6)) pre-training formalizations, we can establish the theoretical connection between these two paradigms. Given optimal models $\mathcal{F}_{\theta^*_{native}}$ and $\mathcal{F}_{\theta^*_{lifted}}$ achieving zero $\mathcal{L}^{pre}_{native}$, $\mathcal{L}^{pre}_{lifted}$ respectively, we have

$$\text{slice}\left(\mathbb{E}_V\left[\mathcal{F}_{\theta^*_{lifted}}(\mathbf{x};G,V)\right]\right) = \mathcal{F}_{\theta^*_{native}}(\mathbf{x};G), \quad (8)$$

where slice denotes taking $t = 0$. This can be easily proven based on the definition of $\boldsymbol{h}_G$ in Eq. (5). This equation indicates that the canonical native-space pretraining essentially learns the expectation of our lifted paradigm, which may lead to information loss and degenerated representations. In contrast, lifted pre-training learns the spanned manifold, where the dynamics $V$ can also serve as the "key" to retrieve corresponding capability learned during pre-training.

### 4.3. Implementation Details

We provide key implementation details here; full configurations can be found in Appendix E.

**Backbone.** GeoPT is architecture-agnostic. We adopt Transolver (Wu et al., 2024), a recent geometry-general neural solver, as our default backbone. We configure three model sizes for scaling experiments: base (8 layers, 3M parameters), large (16 layers, 7M parameters), and huge (32 layers, 15M parameters), all with 256 hidden channels and 32 state tokens. Note that neural simulators typically operate at smaller scales than vision or language models; 15M parameters represents a substantial model for this domain.

**Pre-training data.** We discretize the trajectory in Eq. (4) into 3 steps for a balance between expressiveness and efficiency. For geometric features $\boldsymbol{h}_G(\cdot)$, we use vector distance (Faugeras & Gomes, 2000) to encode global geometry information. All geometries are normalized to unit scale with consistent orientation. For each geometry, we sample 32,768 volume points and 4,096 surface points, with per-point velocities sampled from a bounding ball with radius as 2. For diversity, we generate 100 random dynamics fields per geometry, yielding a million-scale pre-training dataset.

**Computational cost.** The supervision signal in Eq. (6) can be computed efficiently via optimized ray-triangle intersection (Sawhney, 2021). Tracking 36,864 points within one geometry-dynamics sample takes approximately 0.2 seconds on 80 CPU cores, roughly $10^7\times$ faster than industrial-scale CFD simulation (Ashton et al., 2024). We pre-compute all supervision data offline, resulting in a $\sim$5TB dataset, which only takes around 3 days with 80 CPU cores, showing the inherent scalability of geometric pre-training.

**Training parameters.** We pre-train for 200 epochs using AdamW (Loshchilov & Hutter, 2019) with cosine annealing (He et al., 2022). For fine-tuning, we follow Tran-

*Table 1.* Summary of experimental simulations. #Mesh records the size of the discretized meshes for each sample. #Variable records the varied simulation configurations among different samples.

| TYPE | BENCHMARKS | #MESH | #VARIABLE | #SIZE |
|---|---|---|---|---|
| AERO-DYNAMICS | DRIVAERML | $\sim$160M | GEOMETRY | $\sim$6TB |
| | NASA-CRM | $\sim$450K | GEO, SPEED, AOA | $\sim$3GB |
| | AIRCRAFT | $\sim$330K | GEO, SPEED, AOA, SLIP | $\sim$7GB |
| HYDRO- | DTC HULL | $\sim$240K | GEO, YAW ANGLE | $\sim$2GB |
| CRASH | CAR-CRASH | $\sim$1M | IMPACT ANGLE | $\sim$8GB |

solver (Wu et al., 2024) and train for 200 epochs with AdamW and OneCycleLR scheduling (Smith & Topin, 2019) for each downstream physics simulation task.

## 5. Experiments

We extensively evaluate GeoPT in five industrial-scale physics simulation tasks, which involve complex geometries and diverse simulation configurations.

**Benchmarks.** As summarized in Table 1, we examine the model performance on extensive 3D industrial-scale simulations. For aerodynamics, we test Reynolds-Averaged Navier-Stokes (RANS) simulation with DriAverML (Ashton et al., 2024), NASA-CRM (Bekemeyer et al., 2025) and AirCraft (Luo et al., 2025), which are representative high-fidelity data for 3D car and aircraft simulations and require predicting the surface pressure and surrounding wind. As for the hydrodynamics simulation DTCHull, we generate different geometries with ship parameterization (Bagazinski & Ahmed, 2023) and then simulate the ship resistance and wave-making under different geometries and yaw angles with RANS using OpenFOAM (Jasak, 2009), where the model is trained to predict the time-averaged surface pressure and water flow speed. The Car-Crash benchmark is based on an industrial standard model and simulated under different impact angles with OpenRadioss (Altair Engineering Inc., 2026b), where the maximum 2D Von Mises stress for each element during crash is recorded. Although the base geometry is fixed, it involves deformations during crash.

To mimic the industrial practice (Ansys Inc., 2026), we adopt or generate $\sim$100 training samples for each benchmark and test neural simulators on the other 20–50 samples.

**Baselines.** GeoPT mainly focuses on the self-supervised pre-training of neural simulators, which has not been well studied previously. Therefore, we adopt native geometry space pre-training: given position information to predict SDF and vector distance, as baselines. Experimentally, since SDF-based pre-training is much worse than vector distance, we defer the SDF-related experiments to Appendix B. Besides, we also compare with the geometry-conditioned paradigm, where we adopt the VAE encoder from the advanced 3D geometry model Hunyuan3D (Tencent, 2025)

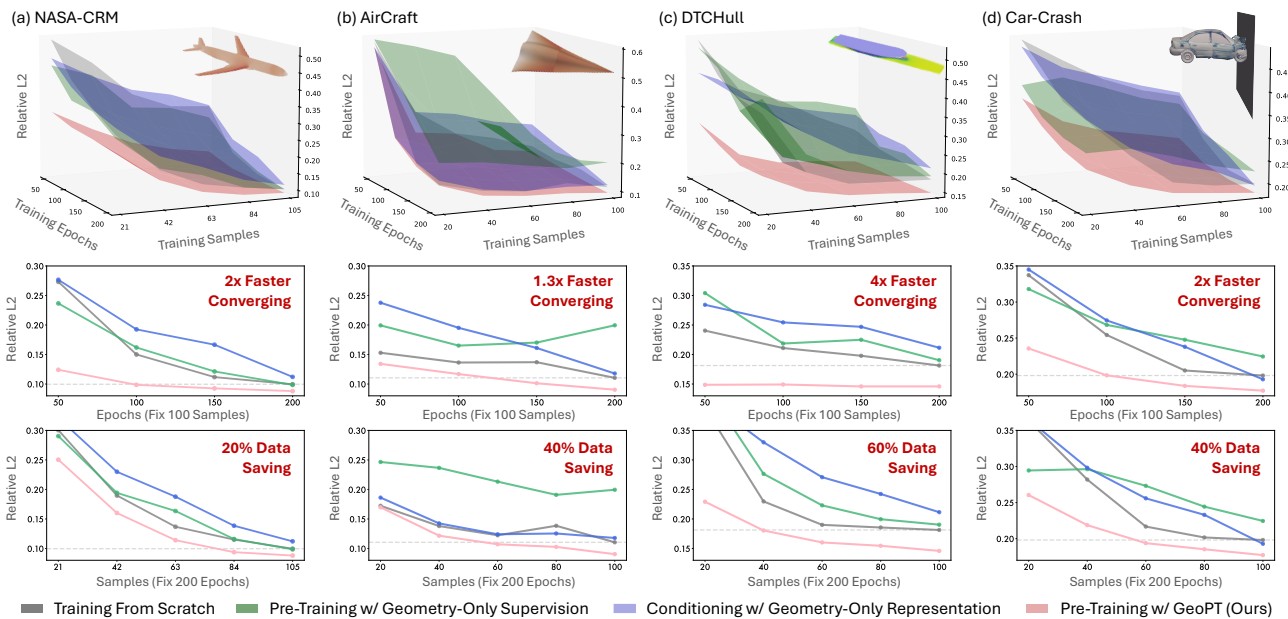

*Figure 5.* Performance comparison across fine-tuning epochs and physics samples. We show detailed curves at 200 epochs and 100 samples for clarity. Here, geometry-only pre-training adopts vector distance, which is better than SDF. See Appendix C for full results.

to extract geometry representations as an auxiliary feature. Additionally, GeoPT is built upon the advanced geometry-general backbone Transolver (Wu et al., 2024). Other Transformer-based simulator backbones, including Galerkin Transformer (2021), GNOT (2023), UPT (2024) and Transolver++ (2025), are also compared. All methods are under the same training strategy to ensure a fair comparison.

### 5.1. Main Results

**Accelerating simulation.** As a supplement to Fig. 1, we further benchmark GeoPT on the other four simulation tasks in Fig. 5, where we can observe that:

*(i) Reducing simulation data requirements.* GeoPT consistently improves a wide range of physics simulations, reducing 20–60% data requirements while reaching performance comparable to full-data training. This improvement is particularly significant in industrial settings, where generating a single training sample may require hours or even days of numerical simulation (Ashton et al., 2024; Bekemeyer et al., 2025). By substantially reducing the amount of simulation data required, GeoPT can alleviate the data bottleneck in AI-enabled industrial workflows (Ansys Inc., 2026).

*(ii) Improving geometry generalization.* GeoPT yields larger improvements in simulations involving a greater diversity of geometries. For instance, GeoPT brings 60% data requirement reduction on DTCHull, where samples exhibit substantial geometric variability in hull curvatures and length-to-beam ratios. In such cases, pre-training on diverse geometries enables GeoPT to better generalize across heterogeneous geometric configurations. In contrast, on NASA-

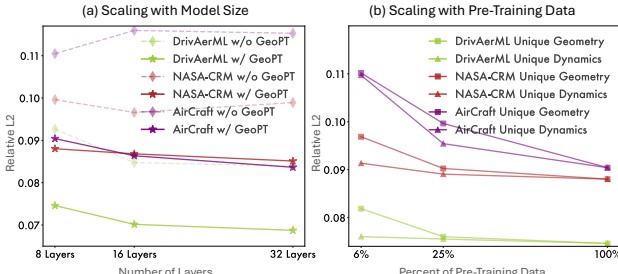

*Figure 6.* GeoPT scaling tests. (a) Gradually increase model layers from 8 to 32 and record the performance change of training from scratch and with GeoPT. (b) Reduce the pre-training diversity of both geometries and dynamics. See Appendix F for full results.

CRM, GeoPT yields a moderate improvement, as the geometric variations among samples are primarily limited to changes in inboard and outboard aileron angles, which only induce slight local deformations in the wing region.

*(iii) Supporting surface-only simulation.* Although our pre-training procedure involves both volume and surface points, GeoPT is also applicable to purely surface-related simulations, such as Car-Crash. By configuring a decayed velocity field on the car surface as the dynamics condition, GeoPT can adapt effectively to this scenario. This flexibility arises from the stochastic nature when generating dynamics-lifted pre-training supervision, which encourages the model to capture a broader range of geometry–physics correlations.

**Scalability.** As a self-supervised model, GeoPT demonstrates favorable scalability in both model and data aspects.

*(i) Model size.* As presented in Fig. 6(a), although the back-

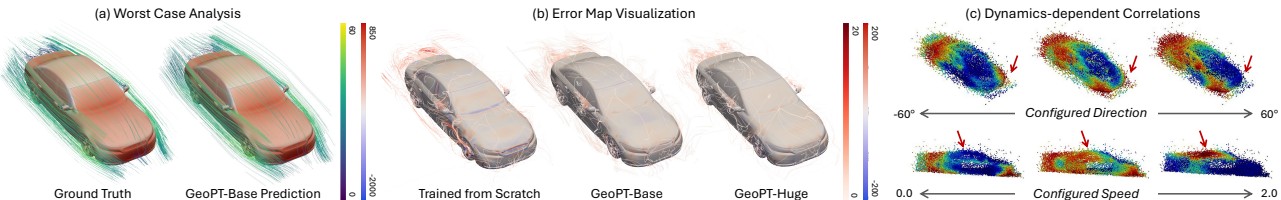

*Figure 7.* Simulation results and learned representations from GeoPT, including (a) visualization of the prediction results with the worst relative L2 performance in DrivAerML, (b) the error map of surface pressure and surrounding velocity, (c) correlations learned by pre-trained GeoPT under varied dynamics information, such as different directions and speeds of $V_S$. See Appendix D for more results.

bone Transolver (Wu et al., 2024) shows nice scalability in sufficient data scenarios, as reported in their paper, it still faces a scaling bottleneck in limited-data industrial simulation, which may be caused by overfitting. In contrast, pre-training with large-scale geometry data can regularize the model hypothesis space to alleviate potential overfitting, thereby consistently benefiting from increasing model size.

*(ii) Data diversity.* In GeoPT, we construct a million-scale pre-training dataset by simulating 100 dynamic trajectories for each unique geometry. To investigate the effect of the pre-training diversity, we separately reduce the number of unique geometries and sampled dynamic trajectories. Results in Fig. 6(b) demonstrate that, compared to dynamic trajectories, the diversity of base geometries is more important to downstream performance. Additionally, the benefit of pre-training dynamics diversity is also task-specific. For example, in DrivAerML with fixed incoming flow, sampling 6% dynamic trajectories can already be comparable to 100 samples, while in AirCraft with varied speed, AoA and sideslip, sampling more dynamic trajectories can significantly increase fine-tune performance, highlighting the potential of GeoPT in handling more complex simulations.

## 5.2. Model Analysis

**Geometry usage.** We provide a detailed ablation of how geometric information is utilized in Fig. 8(a).

*(i) Geometry-only v.s. dynamics-lifted.* Previously, we have extensively discussed the advantage of dynamics-lifting in terms of pre-training. Here, we further explore the representation conditioning usage of GeoPT. For comparison, we adopt geometry representations from the large-scale pre-trained Hunyuan3D (Tencent, 2025) as the baseline, which employs SDF-based geometry reconstruction. While Hunyuan3D representations are effective for accurate geometry reconstruction, they remain insufficient in helping physics simulation (green curve). In contrast, GeoPT employs dynamics-lifted supervision, enabling the model to learn physics-aligned representations (red curve). Thus, GeoPT helps under both pre-training and conditioning, highlighting the essentiality of dynamics lifting in physics simulation.

*(ii) Pre-training v.s. conditioning.* Unlike prior works that

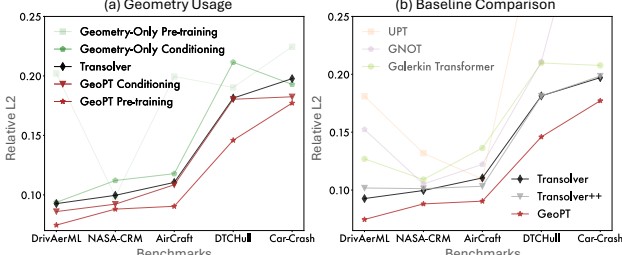

*Figure 8.* (a) Analysis for the geometry usage, including the comparison between geometry-only and dynamics-lifted spaces, as well as pre-training v.s. conditioning. (b) Backbone comparison.

incorporate geometry information as frozen auxiliary features, GeoPT directly leverages geometry to pre-train the physics-learning backbone. As shown in Fig. 8(a), pre-training the backbone yields more substantial performance gains than representation conditioning when effective geometric signals are available. We attribute this to the fact that representation conditioning does not explicitly warm up the physics-learning process, thereby limiting its effectiveness.

**Backbone selection.** In GeoPT, we adopt Transolver (Wu et al., 2024) as the default backbone and compare it with other Transformer-based neural simulators. As shown in Fig. 8(b), under training-from-scratch settings, Transolver consistently outperforms other baseline models in most benchmarks, justifying our choice of backbone. These results further confirm the effectiveness of GeoPT, showing consistent improvements even on advanced benchmarks.

**Worst case study.** We plot the worst prediction case of GeoPT in Fig. 7, where GeoPT still accurately estimates the complex aerodynamics surrounding the car. As demonstrated in Fig. 7(b), compared to training from scratch, GeoPT can improve the prediction accuracy of wake flow and can be further improved by parameter scaling.

**Dynamics-dependent correlations.** Empowered by large-scale pre-training, GeoPT can capture diverse underlying correlations under a proper dynamics prompt. As shown in Fig. 7(c), conditioned on different velocity $V_S$, GeoPT can capture different correlation patterns, such as the inclined correlation under crosswind and the more concentrated correlation under high speed. Especially, under zero speed, our supervision degenerates to static geometry.

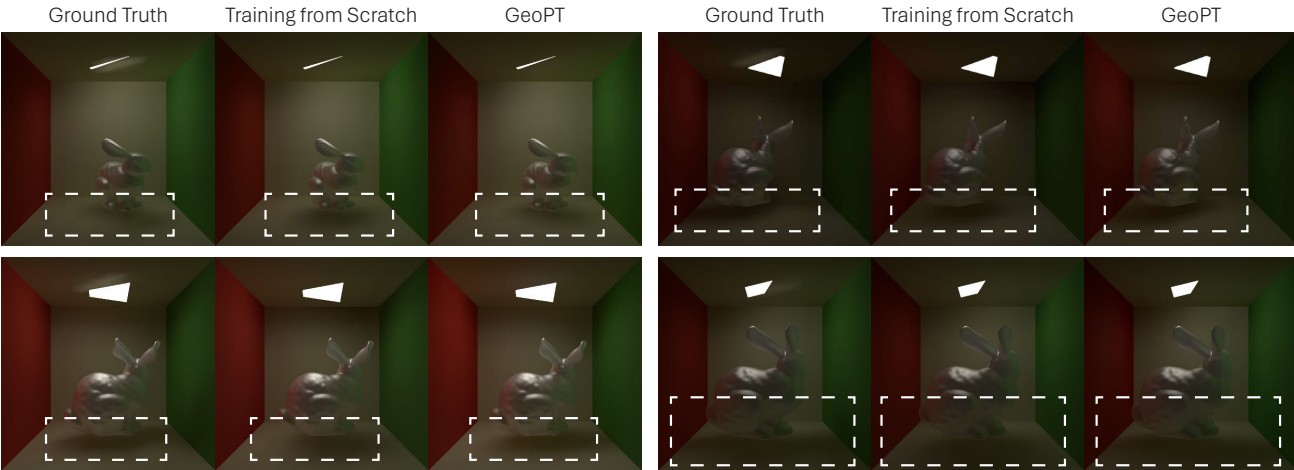

| Ground Truth | Training from Scratch | GeoPT | Ground Truth | Training from Scratch | GeoPT |

*Figure 9.* Experiments of neural radiosity simulation on the Cornell box (Goral et al., 1984) with an unseen Stanford bunny geometry.

### 5.3. Towards Physics Foundation Model

**Generalize to other physics domains.** GeoPT is pre-trained with highly diverse dynamics, endowing it with strong potential to generalize across physics domains. We apply GeoPT to radiosity simulation (Goral et al., 1984), a classical light transport problem, which involves fundamentally different governing physics and geometries from our main experiments. Despite a significant domain shift, GeoPT continues to yield consistent performance improvements, which achieves an MAE of $9.0 \times 10^{-2}$, outperforming training from scratch ($9.7 \times 10^{-2}$). Qualitatively, Fig. 9 reveals that GeoPT captures high-frequency shadow boundaries more accurately, particularly in regions with complex light-geometry interactions. More implementation details of this task can be found in Appendix E.1.

**Pre-training on general-purpose geometries.** As shown in Fig. 6, GeoPT exhibits favorable scalability, suggesting its potential to be further developed into a foundation model (Bommasani et al., 2021) for physics simulation. To further investigate this potential, we enhance GeoPT by increasing the diversity of pre-training geometries. Specifically, we pre-train GeoPT under the following three settings:

*(i) ShapeNet subsets (~10,000 geometries).* We use the car, airplane, and watercraft subsets from ShapeNet, which is also the default pre-training setting in previous experiments.

*(ii) ShapeNet full (~50,000 geometries).* As a general-purpose 3D shape repository, ShapeNet (2015) contains 55 object categories, covering a wide range of shapes such as mobile devices and furniture. Here, we use the full dataset.

*(iii) ShapeNet+Hunyuan3D (~300,000 geometries).* Beyond public 3D assets, advanced 3D generative models can provide an additional source of diverse pre-training geometries. We therefore augment the pre-training data with HY3D-Bench (Hunyuan3D et al., 2026), which contains over 250k geometries generated by Hunyuan3D (2025).

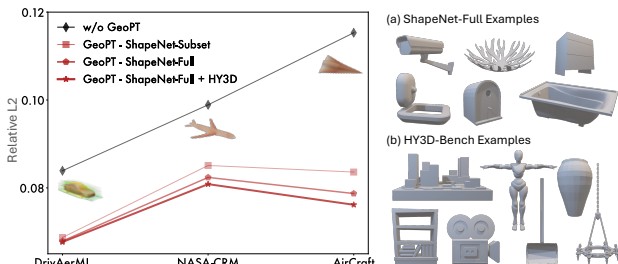

*Figure 10.* Scaling up GeoPT-Huge pre-training with general 3D shape repositories. Left: performance under different pre-training scales. Right: geometry examples from the pre-training datasets.

As shown in Fig. 10, GeoPT can be further improved by increasing the diversity of pre-training geometries, even when these geometries differ substantially from those used during fine-tuning, especially for AirCraft with rarely observed smooth geometries. This result suggests that GeoPT can effectively benefit from large-scale and diverse geometries, providing a practical path toward scaling up pre-training.

Together, the favorable physics and geometry generalizability demonstrated above highlight the potential of GeoPT as a step toward general-purpose physics foundation models.

## 6. Conclusion

This paper presents GeoPT in pursuit of a unified pre-trained model for general physics simulation. Trained solely on geometry data with dynamics-lifted supervision, GeoPT can consistently improve diverse downstream physics, along with a significant reduction in training data requirements and favorable scalability w.r.t. data and model size, demonstrating a possible pathway for scaling neural simulators. As future work, we will further investigate the scaling behavior of GeoPT by continuously expanding the pre-training data and increasing model capacity, as well as validating its effectiveness on broader physical systems.

## Acknowledgements

We would like to thank Hang Zhou and Yuezhou Ma from Tsinghua University for their helpful discussion during the initial exploration of this project.

## Impact Statement

This paper aims to find a scalable way for large-scale pre-training of neural simulators, which is not only valuable for current AI-aided software in industrial design but also poses an intriguing scientific challenge due to the gap between geometry and physics. By newly presenting a lifted self-supervised learning paradigm, we successfully bridge the geometry-physics gap and construct GeoPT solely pre-trained from off-the-shelf geometry data, which reduces 20%–60% training data requirements and achieves $2\times$ faster convergence. Such advantages can significantly accelerate the workflow of AI-aided software. Besides, the lifted learning paradigm also poses a possible way to bridge simple pre-training data and complex downstream tasks, offering a new perspective for self-supervised learning.

Note that this paper mainly focuses on the scientific problem. When developing our approach, we are fully committed to ensuring ethical considerations are taken into account. Thus, we believe there are no potential ethical risks in our work.

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

# A. Theoretical Understanding of GeoPT Pre-Training

This section provides a theoretical interpretation of tracking moving particles in terms of transport equations (also called the collisionless Boltzmann equation or Liouville's equation). We will show that *(i)* tracking a large number of particle trajectories with fixed transport directions is equivalent to solving a transport equation with sticking boundary conditions, and that *(ii)* the resulting dynamics satisfy a natural mass conservation property in the phase space.

**Dynamics process restatement**  Here, we recap the dynamics defined in GeoPT. Let $\Omega \subset \mathbb{R}^C$ denote the open computational domain. In GeoPT pre-training (Eq. (6) of main text), we consider an ensemble of particles initialized with positions $\mathbf{x}_0$ sampled from the computation domain and the geometry boundary and assigned transport directions $\mathbf{v} \sim \text{Unif}(\mathbb{B}^C)$, which remain constant throughout the evolution. Under this setting, each particle follows a free-flight trajectory

$$\mathbf{x}(t) = \mathbf{x}_0 + t\,\mathbf{v}, \qquad t \geq 0. \tag{9}$$

When a particle reaches the geometry boundary $G$, it sticks to the boundary and remains there permanently.

**Phase-space transport formulation**  Since in GeoPT, each tracking particle carries both a position and a transport direction, the most natural continuum description is given in phase space. Let $f(x, v, t)$ denote the phase-space density of particles at position $x$, velocity $v$, and time $t$. Then, under dynamics in Eq. (9), $f$ satisfies the collisionless transport equation, which can be formalized as follows

$$\partial_t f(x, v, t) + v \cdot \nabla_x f(x, v, t) = 0, \qquad (x, v) \in \Omega \times \mathbb{R}^C. \tag{10}$$

It can be proved that, Eq. (10) describes advection in phase space along characteristic curves (Evans, 2010)

$$\dot{\mathbf{x}}(t) = \mathbf{v}, \qquad \dot{\mathbf{v}}(t) = 0, \tag{11}$$

which correspond exactly to the particle trajectories defined in Eq. (9). Therefore, sampling particle trajectories and recording pairs $(x(t), v)$ therefore amounts to sampling characteristic curves of the phase-space transport equation.

Besides, the sticking behavior at the boundary is modeled by allowing particles to transfer from the interior phase space to a boundary-supported phase-space density. Let $f_G(x, v, t)$ denote the phase-space density of particles accumulated on $G$. The flux of particles reaching the boundary is given by $(v \cdot n(x))f(x, v, t)$, where $n(x)$ denotes the outward unit normal. Only particles with $v \cdot n(x) > 0$ reach the boundary. Accordingly, the boundary accumulation satisfies

$$\partial_t f_G(x, v, t) = (v \cdot n(x))_+\, f(x, v, t)\, dv, \qquad x \in G. \tag{12}$$

In summary, Eq. (10) and (12) together define a transport equation with the sticking boundary, which is the continuum limit of GeoPT self-supervision formalized in Eq. (6).

**Proposition A.1** (**Mass conservation**). *The phase-space transport equation in Eq. (10) is conservative. In the absence of sticking, the phase-space flow $(x, v) \mapsto (x + tv, v)$ is divergence-free. With sticking boundaries, mass is not destroyed but transferred from the interior to the boundary. Specifically, the total phase-space mass satisfies (see, e.g., (Evans, 2010))*

$$\frac{d}{dt}\left(\int_\Omega \int_{\mathbb{R}^d} f(x, v, t)\, dv\, dx + \int_G \int_{\mathbb{R}^d} f_G(x, v, t)\, dv\, dS(x)\right) = 0. \tag{13}$$

*Here $dS(x)$ denotes the surface measure on $G = \partial\Omega$. Thus, the dynamics conserve the total number of particles globally, with mass redistributed from the interior to the boundary.*

*Remark* A.2 (**Connect between GeoPT pre-training and solving collisionless transport equation in Eq. (10)**).  In the above analysis, we have demonstrated that the final phase-space density function $f$ of the constructed dynamic process follows the collisionless transport equation. And, in GeoPT, we input the spatial and velocity coordinates of all sampled tracking points into a neural network and supervised the model with L2 loss to predict the dynamic process. Since the dynamic supervision is defined in Eq. (4), which can be seen as a temporal discretization of $f(x, v, t)$, the learning process of GeoPT is actually to estimate $f(x, v, t)$ from the corresponding spatial and velocity coordinates.

As stated in proposition A.1, the flow density $f$ that is used as a self-supervision signal obeys mass conservation under all kinds of velocity settings. During the pre-training process, we randomly sample geometries $g$ and velocity fields $\mathbf{v}$ as the model input. Beyond ensuring data diversity, the above-described pre-training will enforce the model output maintain the mass conservation law under all kinds of geometries and dynamics.

*Remark* A.3 (**Generalizability of theoretical framework**). The phase-space transport equation captures a canonical conservation-law structure: mass evolves by advection along characteristics, with boundary flux accounting. Many physical models can be viewed as augmentations of this structure. For example, the Boltzmann equation adds a collision operator to the streaming term, and fluid equations such as Navier-Stokes couple mass conservation with additional momentum and constitutive relations. Consequently, while our pre-training signal is derived from a simplified transport process, it instills an inductive bias toward characteristic-driven correlations and boundary interactions, which are shared components across a broad class of continuum and kinetic systems. We therefore expect the pre-trained representation to transfer to downstream PDE solvers after task-specific adaptation, rather than guaranteeing universal generalization by itself.

## B. Pre-Training Investigation

This section provides the investigation of GeoPT's pre-training configurations in the dataset, dynamics and supervision.

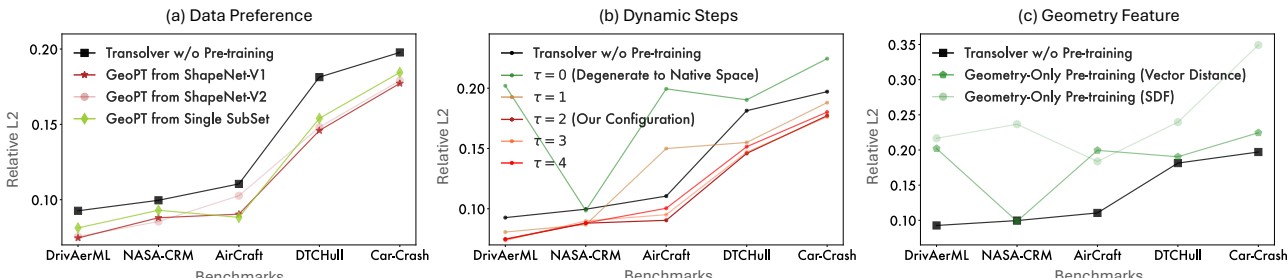

*Figure 11.* Ablations for the pre-training choices in GeoPT, including the comparison among (a) pre-training with a single subset or mixed data from ShapeNet-V1, -V2, (b) discretizing the dynamics into various step numbers $\tau$, and (c) pre-training with different geometry information, such as SDF and vector distance. All the experiments are based on GeoPT-Based under the full-data-full-training setting.

**Ablation 1: single v.s. mixed geometry dataset** In GeoPT, we attempt to build a unified pre-trained model for all types of physics simulations. Thus, the default setting in GeoPT is to use a mix of cars, airplanes, and watercraft as the pre-training dataset. However, it is also possible to adopt GeoPT on a case-by-case basis, such as pre-training only on the car subset for DrivAerML and Car-Crash simulation, only on the airplane subset for NASA-CRM and AirCraft, and only on the watercraft subset for DTCHull, which can better align with the geometry in the downstream task. We have also compared GeoPT with the single-subset pre-training setting in Fig. 11(a) red v.s. green curves. It is observed that in most tasks, a more diverse pre-training brings better performance, highlighting the value of unified pre-training. Only for AirCraft, single-subset pre-training is slightly better. Notably, this task can be viewed as a corner case, which contains quite different geometries w.r.t. the pre-training geometries (Fig. 17), thereby involving cars or watercraft that are far away from AirCraft geometries in pre-training may cause distraction. One possible solution is to keep enlarging the diversity of pre-training geometries.

**Ablation 2: ShapeNet-V1 (low quality but high diversity) v.s. ShapeNet-V2 (high quality but low diversity)** In the official configuration of GeoPT, we adopt the ShapeNet-V1 (Chang et al., 2015) for pre-training, which contains 13,463 geometries in total for car, watercraft and airplane categories. However, geometries in the initial version of ShapeNet may contain incorrect normals and non-aligned orientations. ShapeNet-V2 is an updated version with manually corrected meshes, normals, and normalized orientations, but it contains only 9,515 geometries in the three industrial-related categories. In Fig. 11(a), we compare the performance of GeoPT pretrained from ShapeNet-V1 and -V2. It is observed that in most benchmarks, ShapeNet-V1 pre-training brings better performance, even though it is based on geometries without careful quality control. This finding also highlights the importance of geometry diversity, encouraging the wide collocation of 3D geometries, where the advanced 3D generation models can be a good data source (Tencent, 2025).

**Ablation 3: step number in dynamics configuration** As described before, we discretize the dynamic process into $(\tau + 1)$ steps, which generates the dynamic trajectory supervision $\boldsymbol{h}_G(\mathbf{x}_{0:\tau}) \in \mathbb{R}^{(\tau+1)\times C}$. As presented in Fig. 11(b), $\tau = 0$ corresponds to the degeneration scenario, which is still the static geometry supervision, thereby cannot bring benefits to physics simulation. One step forward, namely $\tau = 1$, is able to bring a significant promotion, highlighting the necessity of dynamics representation. In general, $\tau = 2$ achieves a balanced performance across various physics simulation tasks, which is also set as our default configuration. Besides, adding dynamic steps will not consistently bring benefits since the accumulated discretization error, which is why $\tau = 3, 4$ can only surpass the default configuration in part of the benchmarks. It is also worth noting that increasing $\tau$ will also cause more computation costs. Thus, $\tau = 2$ is a well-verified choice.

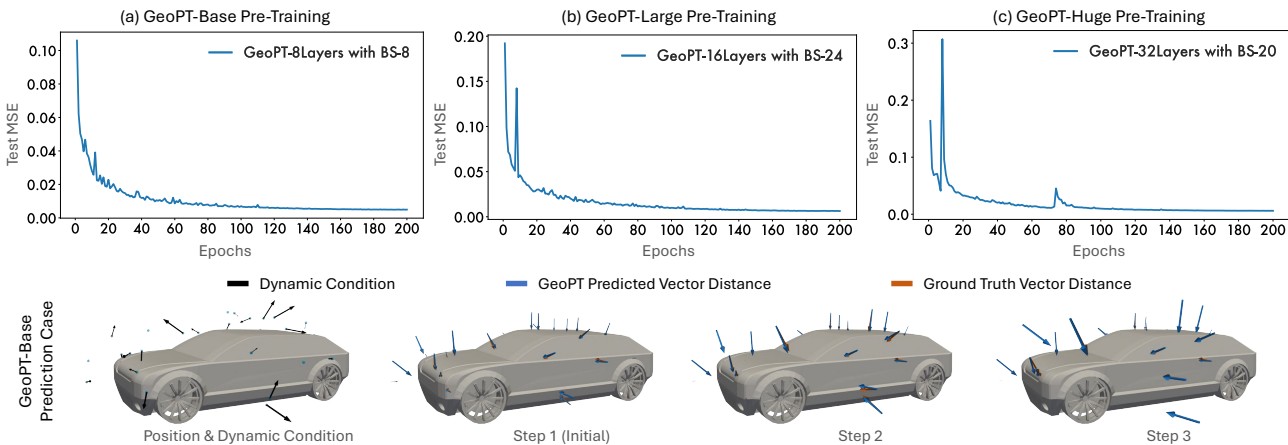

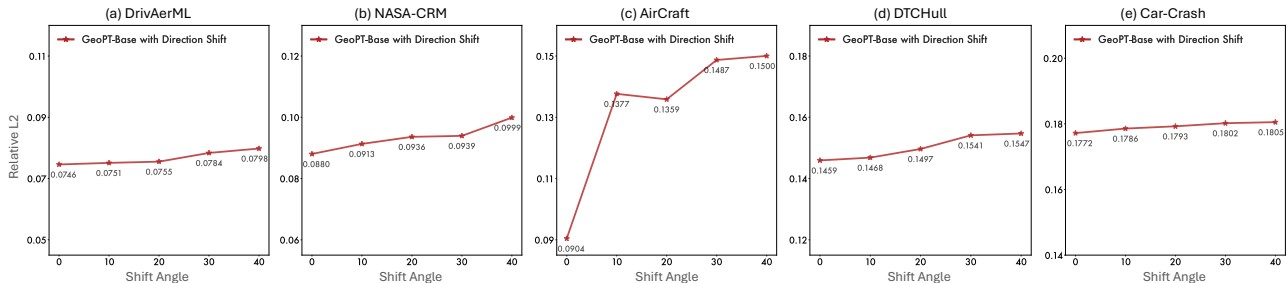

*Figure 12.* The pre-training process of GeoPT. (a) We plot the test MSE change during 200 pre-training epochs, which is calculated on 300 leave-out geometries. The initial unstable stage is caused by *learning rate warm up*. (b) Prediction case of the pre-trained GeoPT-Base. For clarity, we only plot 35 points under 3 dynamic steps, where the prediction is in blue and the ground truth is in orange.

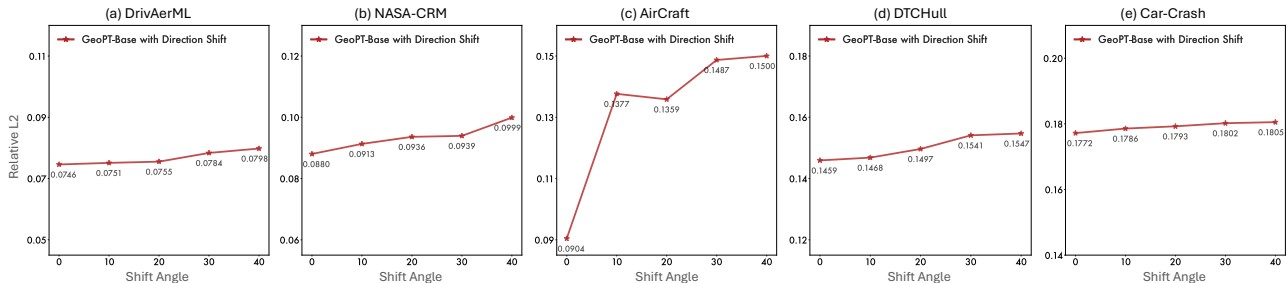

*Figure 13.* GeoPT-Base performance change w.r.t. the direction *shifted* (by $0°$ to $40°$) from the correct configuration. Zero shift refers to configuring the dynamics field direction along the incoming flow or impact angle, which is our default setting. For clarity, we adopt the same $y$-axis range of 0.75 among different benchmarks in visualization. All the experiments are under the full-data-full-training setting.

**Ablation 4: vector distance v.s. SDF in choosing geometry feature** In the main text, we have compared with pre-training with vector-distance-based geometry-only supervision. It is also applicable to adopt SDF as the supervision. However, as shown in Fig. 11(c), SDF supervision is generally worse than vector distance. This result may be because vector distance is more informative than SDF for each point, which not only contains distance information but also involves the direction. Based on this result, in GeoPT, we also adopt vector distance as $\boldsymbol{h}_G(\cdot)$ in Eq. (5) to encode the geometry information.

**Pre-training visualization** To provide an intuitive understanding of GeoPT pre-training, we plot the test loss curve in Fig. 12, where GeoPT with different sizes all converge smoothly. Specifically, the relatively unstable curve for the early stage during pre-training is caused by learning rate warm-up. Besides, Fig. 12 also presents the prediction results for self-supervised dynamic processes. Specifically, the initial geometry information and dynamics field are input to GeoPT and the model is expected to predict the vector distance w.r.t. the geometry surface for the future three steps. We can find that the pre-trained GeoPT can precisely predict the future dynamics, indicating sufficient optimization in pre-training.

## C. Fine-Tuning Investigation

This section provides the analysis for the usage and special features of GeoPT during fine-tuning.

**Fine-tuning configuration for particle velocity** As described in the main text, we need to parameterize the simulation setting into particle velocity $V_S$ for GeoPT fine-tuning. Specifically, we need to determine both the direction and norm:

- As for the direction of the particle velocity $V_S$, the configuration is quite certain in practice, which can be directly calculated based on the direction of incoming flow in aerodynamics (*e.g.*, yaw angle in DTCHull or angle of attack and sideslip in AirCraft) or force direction in solid mechanics (*e.g.*, impact angle of Car-Crash).

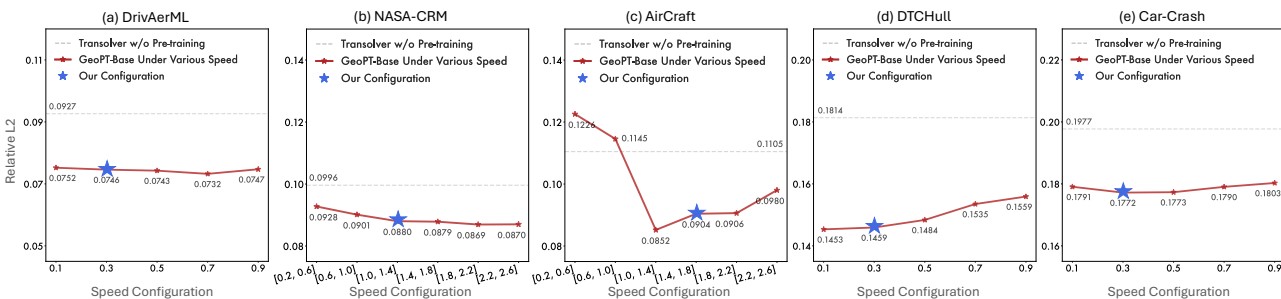

*Figure 14.* GeoPT-Base performance change w.r.t. the *configured velocity norm in $V_S$*, which is sampled from $[0, 2]$ during pre-training. For real-world low-speed cases, such as car and watercraft simulations, we explore the choices within less than 1.0. As for high-speed cases, such as aircraft, we explore a larger range $[0.2, 2.6]$. Since both NASA-CRM and AirCraft involve varying simulation speeds, we normalize the real-world configuration into different intervals. All the experiments are under the full-data-full-training setting.

- As for the norm of the particle velocity $V_S$, corresponding to speed, it needs more calibrations, since the pre-training is based on the normalized geometry and normalized moving speed, while the simulation speed is a real-world value. Therefore, a correspondence between the normalized geometry space and the real-world physics space is expected.

In our experiments, we do not elaborately tune the configuration of $V_S$, especially for its norm. To give an intuitive understanding and practical recipe for fine-tuning configurations, we provide a detailed analysis here.

*(i) Performance under direction shift.* Fig. 13 demonstrates that directly adopting the incoming flow or impact direction from downstream simulation settings achieves the best performance, which corresponds to the zero shift setting. Although globally shifting the direction configuration by a fixed value can also ensure a quantitatively distinguishable condition value among different samples for the model, it will cause performance degradation since it will inform the pre-trained with incorrect correlations, as visualized in Fig. 7(c). Therefore, the performance drop gets more serious under a larger direction shift. Specifically, the performance drop is more significant for high-speed scenarios, such as NASA-CRM and AirCraft, because their high-speed configuration will magnify the effect of incorrect configuration of direction.

*(ii) Performance under various speed configurations.* To ensure a precise alignment between pre-training supervision and fine-tuning configuration, we need to consider all the simulation settings, such as object length, flow viscosity, and real-world speed, to determine the norm of $V_S$. However, in practice, we find that a roughly reasonable speed configuration is sufficient to deliver a fair performance. In our experiments, we set the norm of elements in $V_S$ as 0.3 for all low-speed scenarios, including DrivAerML, DTCHull and Car-Crash[2] and normalize the real-world speed in NASA-CRM and AirCraft into an interval with larger values, which is $[1.0, 1.4]$ and $[1.4, 1.8]$, respectively. In general, as presented in Fig. 14, the low-speed scenarios prefer smaller speed configurations and the high-speed task requires a $V_S$ with a larger norm, as the latter usually corresponds to more concentrated physical states, which can be "prompted" by a larger speed configuration (Fig. 7(c)).

Based on the above analysis, we summarize the following usage recipe for GeoPT.

> **Configuration recipe for dynamics conditions** $V_S$. *In general, the direction of the velocity field should be set following the simulation settings, while the norm of velocity can be set within* $[0.1, 1.0]$ *for commonly seen low-speed simulations, such as driving cars or ships with speed less than 100 m/s, and be set within* $[1.0, 2.0]$ *for high-speed tasks.*

*Remark* C.1 (**Distinguishable configurations**). It is worth noticing that without pre-training, the dynamics configuration only needs to be distinguishable to reflect the exact simulation configurations since the model will automatically learn to project the input into deep representations. In GeoPT, we pursue a suitable dynamics configuration to align the pre-training model and the downstream task, where different dynamics configurations activate different correlation patterns (Fig. 7(c)), while the effective range of configuration is quite flexible as the model can be further optimized during finetuning.

**Fine-tune loss curves** Here, we also plot the test loss during the fine-tuning process in Fig. 15, where GeoPT can consistently improve the convergence performance and help to accelerate the training stability, especially for the early stage, such as DrivAerML and DTCHull. Note that the physics simulation task evaluated in this paper is quite challenging, which

---

[2]In this paper, we configure the dynamics field in Car-Crash with a linearly decayed norm. Here, we only consider the maximum speed value. If we adopt the same speed value for all elements, there will be a slight performance drop, where the relative L2 is 0.1783.

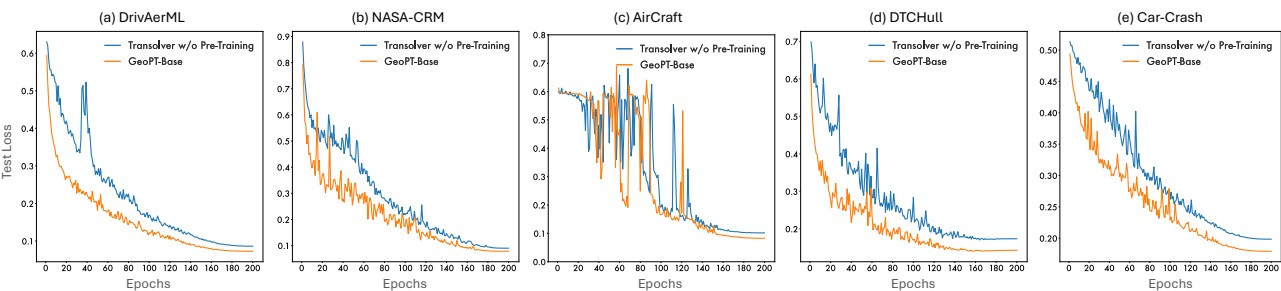

*Figure 15.* Test loss (relative L2) change during fine-tuning. Note that these test losses are calculated from the downsampled physics field to ensure training efficiency, which is proportional to the final performance, but may be shifted w.r.t. the full mesh results.

requires the model to infer the whole physics field solely from geometry information and simulation settings. Therefore, the training processes of all benchmarks are not very stable in the early stage, but will converge smoothly in the end.

As for the AirCraft benchmark that involves a serious out-of-domain generalization challenge due to the training-test geometry gap, the training process is not very stable as presented in Fig. 15(c), where we have also tried $10\times$ smaller learning rates, but still cannot ensure a smooth convergence, indicating the learning difficulty of this task. Still, in this task, GeoPT can improve the final convergence performance, highlighting its effectiveness.

## D. Showcases

As a supplement to Fig. 7, we plot the showcases from the other four benchmarks in Fig. 16-19.

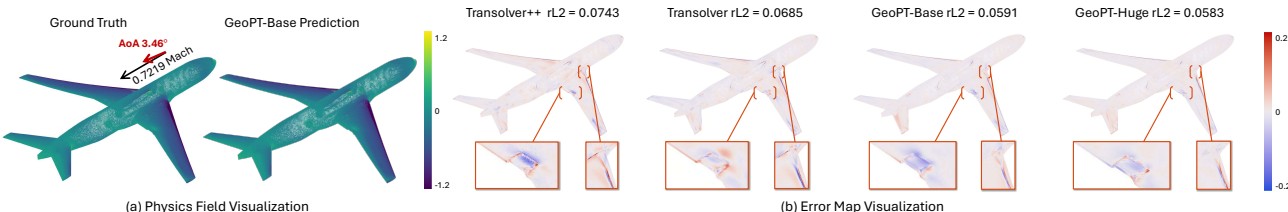

*Figure 16.* Showcase of NASA-CRM. (a) The pressure coefficient field of the airplane flying under 0.7219 Mach and 3.46° angle of attack. (b) Error map and the relative L2 of different models for this case. For clarity, we zoom in on high-error zones.

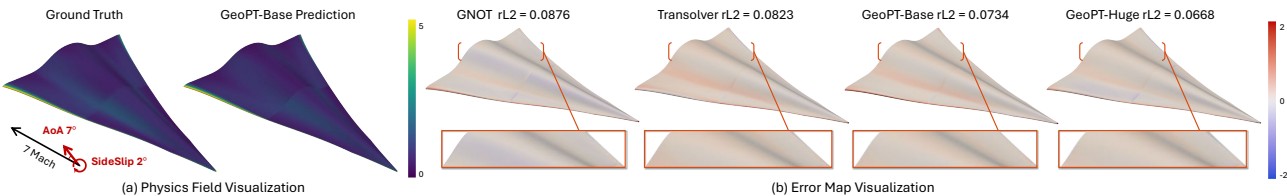

*Figure 17.* Showcase of AirCraft. We plot the Z-force coefficient field for aircraft flying under 7 Mach, 7° angle of attack and 2° sideslip.

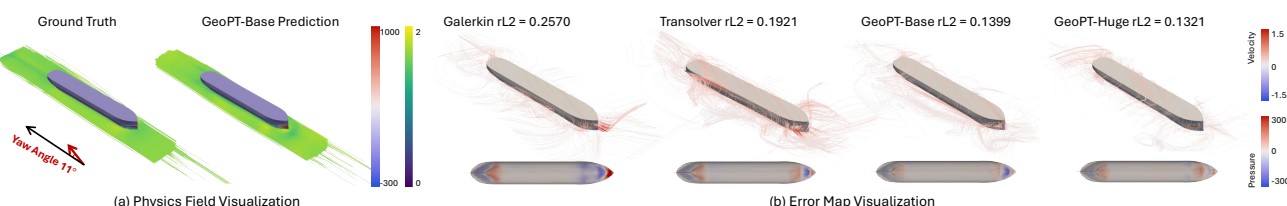

*Figure 18.* Showcase of DTCHull. (a) We plot the hydrostatic pressure–corrected pressure and surrounding velocity streamlines for a ship moving under 11° yaw angle. (b) We highlight the surrounding velocity error in the streamline and the pressure error in underside view.

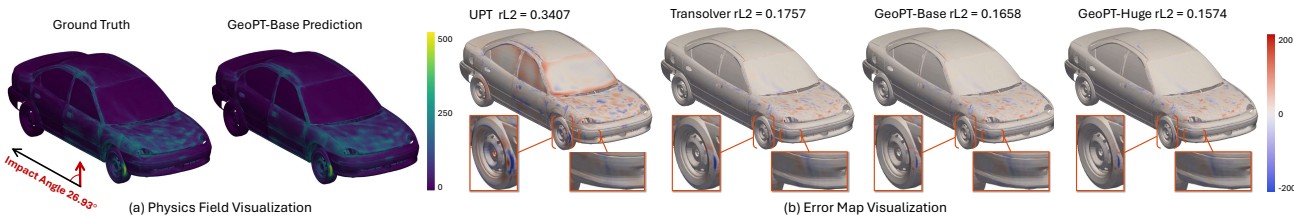

*Figure 19.* Showcase of Car-Crash. We plot the element-wise maximum 2D Von Mises stress during the crash with $26.93°$ impact angle.

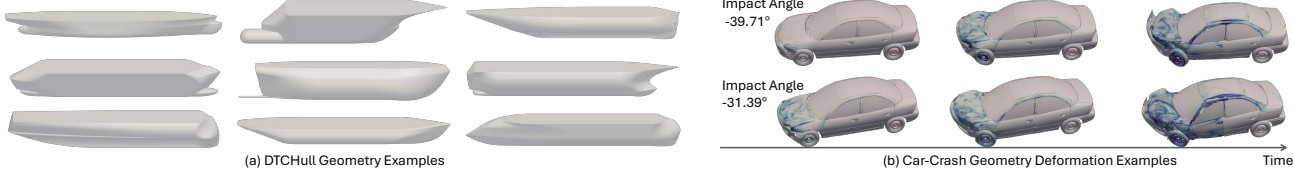

*Figure 20.* Examples from DTCHull and Car-Crash benchmarks, which involve diverse geometries and deformations, respectively.

As presented in the first subfigure, GeoPT can accurately predict the physics field of challenging simulation tasks, which involve complex geometries and diverse simulation settings. For example, in AirCraft, the model needs to predict six aerodynamic forces and moments simultaneously and this task contains four variables for different cases: geometry, speed, angle of attack and sideslip, making the prediction extremely challenging, while GeoPT can bring over 10% improvement over Transolver trained from scratch and consistently benefits from model scaling.

As for DTCHull and Car-Crash, both tasks involve intricate physical interactions, where the former requires the model to predict the water-air two-phase interactions and the latter needs to predict the maximum stress during structure deformation caused by the crash. Especially for Car-Crash, the stress field can be discontinuous due to the fracture. In these two difficult tasks, lifted geometric pre-training can still improve the performance and significantly surpass the baselines.

## E. Implementation Details

This section will introduce implementation details for benchmark and data generation, training and baseline implementations.

### E.1. Benchmarks

We experiment with five industrial design tasks and one radiosity task to examine the model generalizability, where DrivAerML (Ashton et al., 2024), NASA-CRM (Bekemeyer et al., 2025) and AirCraft (Luo et al., 2025) are adopted from the previous work. We also newly simulate three benchmarks, including DTCHull, Car-Crash and Radiosity.

*Table 2.* Summary of experimental simulations. #Mesh records the size of the discretized meshes for each sample. #Variable records the varied simulation configurations among different samples. #Train and #Test represent the number of training and test samples.

| TYPE | BENCHMARKS | #MESH | #VARIABLE | #SIZE | #TRAIN | #TEST | OUTPUT PHYSICS |
|---|---|---|---|---|---|---|---|
| AERO-DYNAMICS | DRIVAERML | ~160M | GEOMETRY | ~6TB | 100 | 20 | PRESSURE & VELOCITY |
| | NASA-CRM | ~450K | GEO, SPEED, AOA | ~3GB | 105 | 44 | PRESSURE COEFFICIENT |
| | AIRCRAFT | ~330K | GEO, SPEED, AOA, SIDESLIP | ~7GB | 100 | 50 | AERODYNAMIC SIX COMPONENTS |
| HYDRO- | DTCHULL | ~240K | GEO, YAW ANGLE | ~2GB | 100 | 20 | TIME-AVG PRESSURE & VELOCITY |
| CRASH | CAR-CRASH | ~1M | IMPACT ANGLE | ~8GB | 100 | 30 | TIME-MAX 2D VON MISES STRESS |

**DTCHull** This task is to simulate the ship resistance and wave-making process. First, we generate 130 different ship geometries by changing the hull parameterization script from (Bagazinski & Ahmed, 2023), which makes it easy to produce diverse geometries (Fig. 20). Then, we simulate the two-phase incompressible free-surface flow of water and air surrounding different ship geometries for 50 seconds using the Volume-of-Fluid (VoF) multiphase solver in OpenFOAM (Jasak, 2009), with constant fluid properties ($\rho_{\text{water}} = 998.8$, $\nu_{\text{water}} = 1.09 \times 10^{-6}$; $\rho_{\text{air}} = 1$, $\nu_{\text{air}} = 1.48 \times 10^{-5}$). The air–water interface is initialized as a flat surface located at 0.244 above the zero $x$–$y$ plane, and we neglect surface tension. Turbulence is modeled with Reynolds-averaged Navier-Stokes (RANS) using the $k - \omega$ Shear Stress Transport model closure.

We set the water velocity as a normalized value of 1.668 and the air velocity as zero at initialization, which roughly corresponds to a 12 m/s moving scenario in the real-world scale. For each case, we randomly sample a yaw angle from $[-10°, 10°]$ to mimic real-world diversity. Finally, we average the hydrostatic pressure–corrected pressure and surrounding velocity over the last 20 seconds as the target, when the dynamics are relatively stable.

**Car-Crash**    This task is to simulate high-speed impact dynamics with large deformations, contact interactions, and potential material failure jointly determine the evolving vehicle geometry. We adopt the National Crash Analysis Center Neon model, a widely used full-vehicle crash benchmark with detailed part-level meshing and heterogeneous material assignments. Each simulation is carried out in OpenRadioss (Altair Engineering Inc., 2026b) using a 3D Lagrangian explicit finite-element formulation of transient momentum balance, where external loads and penalty-based contact forces are assembled at nodes and internal forces are computed element-by-element from stresses. At the element level, stresses are advanced by integrating the constitutive (material) law, including elastic/plastic response and, when enabled, damage/failure/erosion, driven by the local deformation history.

For each run, we simulate 17 seconds of dynamics under a rigid-impact configuration, with an impact angle sampled uniformly from $[-45°, 45°]$, and record the maximum von Mises equivalent stress attained by each element over the entire trajectory. As shown in Fig. 20(b), although the initial geometry is fixed, varying the impact angle induces substantially different contact sequences and load paths, leading to distinct deformation and final geometry.

**Radiosity**    To evaluate whether GeoPT transfers to physical regimes beyond those evaluated in our main experiments, we apply it to radiosity simulation (Goral et al., 1984), a classical light transport problem that computes global illumination by modeling diffuse inter-reflections between surfaces. This problem has fundamentally different governing physics from the fluid and solid mechanics tasks in our main experiments. We construct a dataset of 200 radiosity renderings in a Cornell box scene (Goral et al., 1984) with a Stanford bunny placed at varying positions and scales, illuminated by light sources with different intensities, sizes, and directions. We parameterize the dynamics conditions as the light propagation direction, analogous to the flow direction in aerodynamics. We finetune GeoPT on 160 training samples and evaluate on 40 held-out test samples. Notably, neither the Cornell box geometry nor any light transport physics was seen during pre-training. This benchmark examines GeoPT's potential as a general-purpose prior for diverse simulation tasks.

### E.2. Experiment Configuration

The detailed configurations can be found in Table 3. All the experiments are conducted in PyTorch (Paszke et al., 2019) on NVIDIA A100 40GB GPU. Gradient checkpointing (Chen et al., 2016) is used in the pre-training of GeoPT-large and -huge.

*Table 3.* Configurations for GeoPT experiments. "B, L, H" represent the base, large and huge models.

*(a)* Self-supervision data configuration.

| TERMS | CONFIGURE |
|---|---|
| *Used Geometry Data $\mathcal{G}$* | |
| CAR SUBSET | 7,479 SHAPES |
| AIRPLANE SUBSET | 4,045 SHAPES |
| WATERCRAFT SUBSET | 1,939 SHAPES |
| *Geometry bounding box to sampling tracking particles* | |
| CAR | $[-3.0, 4.2] \times [-1.5, 1.5] \times [0.0, 2.5]$ |
| AIRPLANE | $[-3.0, 4.5] \times [-2.8, 2.8] \times [0.0, 2.5]$ |
| WATERCRAFT | $[-4.0, 5.0] \times [-1.5, 1.5] \times [-1.0, 1.0]$ |
| *Tracking particles $N$* | |
| VOLUME | 32,768 |
| SURFACE | 4,096 |
| *Dynamic process configuration* | |
| NUMBER OF STEPS $\tau$ | 2 |
| MAXIMUM VELOCITY $v_{\text{MAX}}$ | 2 |
| DYNAMICS PER GEOMETRY | 100 |
| **PRETRAINING SAMPLES** | 1,346,300 SAMPLES ($\sim$ 5TB) |

*(b)* Training configurations.

| TERMS | CONFIGURE |
|---|---|
| *Pre-Training Stage* | |
| OPTIMIZER | ADMAW (2019) |
| BASE LEARNING RATE | 0.001 |
| LEARNING RATE SCHEDULE | COSINE |
| BATCH SIZE | 8 (B), 24 (L), 20 (H) |
| WARMUP EPOCHS | 10 |
| TOTAL EPOCHS | 200 |
| LOSS FUNCTION | L2 LOSS |
| **APPROX GPU HOURS** | 144 (B), 360 (L), 576 (H) |
| *Fine-Tuning Stage* | |
| OPTIMIZER | ADMAW (2019) |
| BASE LEARNING RATE | 0.001 |
| LEARNING RATE SCHEDULE | ONECYCLELR |
| BATCH SIZE | 1 |
| WARMUP EPOCHS | PYTORCH DEFAULT |
| TOTAL EPOCHS | 200 |
| LOSS FUNCTION | RELATIVE L2 LOSS |
| **APPROX GPU HOURS** | 3 (B), 6 (L), 10 (H) |

**Pre-training** During pre-training, we only go through all the unique geometries in each epoch, which does not use all 100 dynamic trajectories for each geometry but randomly samples one from them within each epoch.

**Fine-tuning** For all methods, due to GPU memory limitations when processing extremely large geometries, we split the input mesh into several subsets with $50,000$-$100,000$ mesh points per subset and train the simulator on these downsampled geometries and physics fields. Regarding inference, we utilize the geometry-general property of Transformer solvers and directly infer the entire mesh at one forward pass, which is slightly more effective than separately inferring and then concatenating results in our experiments. Especially for the DrivAerML benchmark (Ashton et al., 2024) that contains 160M mesh points per sample, directly inferring the full mesh will cause out-of-memory. Therefore, in this benchmark, we split the surface mesh into 20 subsets and the volume mesh into 400 subsets at the beginning, and infer these subsets sequentially, where one volume subset is paired with one surface subset for inference.

### E.3. Self-Supervision Data

Here, we provide the implementation details of self-supervision data generation in Algorithm 1. All three loops for geometry, dynamics and moving steps in the following algorithm can be executed in parallel.

---

**Algorithm 1** Self-supervision data generation in GeoPT

---

**Input Data:** Geometry dataset $\mathcal{G}$
**Input Config:** Number of tracking points $N$, number of time steps $\tau$, maximum velocity $v_{\max}$, number of velocity fields per geometry $N_{\mathrm{dyn}}$
**Output:** Pre-training dataset $\mathcal{D}$ with $(N_{\mathrm{dyn}} \times |\mathcal{G}|)$ samples
Initialize dataset $\mathcal{D} \leftarrow \emptyset$
**for** $G$ in $\mathcal{G}$ **do**
  // **Step 1: Normalize geometry**
  $G \leftarrow \mathrm{Rotate}(G)$ // Rotate geometry to align front face to $-x$ direction.
  $G \leftarrow \mathrm{Shift}(G)$ // Zero-center $x$–$y$ coordinates; place bottom on $x$-$y$ plane.
  $G \leftarrow \mathrm{Scale}(G)$ // Scale to unified $x$-axis length as 5.
  Build FCPW scene (Sawhney, 2021) for $G$. Blue symbols indicate FCPW-accelerated operations.
  // **Step 2: Sample query positions**
  $\{\mathbf{x}\} \leftarrow \mathrm{Sample}(\Omega_G \cup \partial G)$ // Sample from bounding box $\Omega_G$ and surface $\partial G$.
  $\{\mathbf{x}\} \leftarrow \mathrm{Outside}(\{\mathbf{x}\}, G)$ // Remove points inside $G$; retain $N$ tracking points.
  **for** $i$ in $\{1, \ldots, N_{\mathrm{dyn}}\}$ **do**
    // **Step 3: Sample synthetic velocity field**
    $\{\mathbf{v}\} \leftarrow \{\mathrm{Sample}(\mathbb{B}^C)\}$ where $\mathbb{B}^C = \{\mathbf{v} \in \mathbb{R}^C : \|\mathbf{v}\|_2 \leq v_{\max}\}$ // Per-point i.i.d. sampling.
    // **Step 4: Compute feature trajectory following Eq. (4) and Eq. (5)**
    Initialize trajectory $\boldsymbol{h}_G(\mathbf{x}_{0:\tau}) \leftarrow \emptyset$
    $\{\mathbf{x}_0\} \leftarrow \{\mathbf{x}\}$
    **for** $t$ in $\{0, \ldots, \tau\}$ **do**
      $\boldsymbol{h}_G(\{\mathbf{x}_t\}) \leftarrow \mathrm{VectorDistance}(\{\mathbf{x}_t\}, G)$ // Compute geometric features.
      Append $\boldsymbol{h}_G(\{\mathbf{x}_t\})$ to $\boldsymbol{h}_G(\{\mathbf{x}_{0:\tau}\})$
      $\{\mathbf{x}_{t+1}\} \leftarrow \mathrm{Evolve}(\{\mathbf{x}_t\}, \{\mathbf{v}\}, G)$ // Update position: $\mathbf{x}_{t+1} = \mathbf{x}_t + \mathbf{v} \cdot \mathbb{1}_G(\mathbf{x}_t)$.
    **end for**
    Add $\{(\{\mathbf{x}\}, \{\mathbf{v}\}, G), \boldsymbol{h}_G(\mathbf{x}_{0:\tau})\}$ to $\mathcal{D}$
  **end for**
**end for**
**Return** $D = \{(\{\mathbf{x}\}, \{\mathbf{v}\}, G), \boldsymbol{h}_G(\mathbf{x}_{0:\tau})\}$

---

### E.4. Baselines

Here, we detail our implementation for baselines, including backbone selection and geometry usage.

**Backbone selection** We adopt the implementation of Galerkin Transformer (Cao, 2021), GNOT (Hao et al., 2023) and Transolver (Wu et al., 2024) from the open-sourced repository Neural-Solver-Library[3], which provides high-quality

---

[3]https://github.com/thuml/Neural-Solver-Library

implementations of various neural PDE solvers and has been verified by the authors of these papers. As for Transolver++ (Luo et al., 2025) and UPT (Alkin et al., 2024), we adopt their official code. For backbone comparison, we configure all the Transformer backbones as 8 layers with 256 hidden channels. Besides, we also adopt the same simulation parameterization method proposed by GeoPT to incorporate condition information into these baselines.

**Geometry usage**  Since prior research does not directly study geometric pre-training for physics, we compare GeoPT with two reasonable baselines. Here are details for these two geometry usage baselines.

*(i) Geometry-only pre-training.* In this type of baseline, we only adopt the geometry-only feature as supervision. Specifically, we train the model to predict SDF or vector distance (Faugeras & Gomes, 2000) based on given positions.

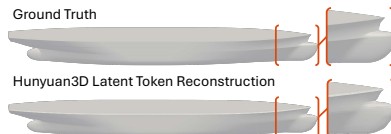

*(ii) Geometry-only conditioning.* The prior work (Zhang et al., 2026) has explored using frozen geometry representation as the auxiliary information for PDE solving. However, the 3D model used in their work is limited to 3D inductors, which cannot be used in complex industrial designs tested in our paper. Thus, we adopt

*Figure 21.* Reconstruction examples of latent tokens extracted by Hunyuan3D.

the advanced geometry model Hunyuan3D (Tencent, 2025) to extract the static geometry representation for comparison. Specifically, we adopt the pre-trained VAE model[4] encoder, which can receive a set of points sampled from a mesh geometry as input and encode it into 3,072 geometry tokens with 64 hidden channels. Then we integrate the extracted token sequence into Transolver based on an additional cross-attention layer to fuse the geometry tokens into input mesh representations, which is also the default usage in (Zhang et al., 2026). As shown in Fig. 21, the geometry representation learned by Hunyuan3D can precisely capture the detailed geometry information, thereby enabling accurate geometry reconstruction. However, as presented in the main results (Fig. 5), the static geometry representation cannot help the physics learning, even though it has already precisely encoded the geometry information.

## F. Full Results

**Extension to time-dependent simulation**  In the main text, we only present the model performance on steady state simulation. Here, we applied GeoPT to the transient simulation DTCHull, which is the DTCHull before time-aggregation and requires simulating the next transient field based on the current observation. Unlike steady-state, transient field correlation is more short-term. Thus, we configure the velocity norm $\|V_S\|$ to be $5\times$ smaller than the steady-state value for fine-tuning, where GeoPT still yields benefits compared to training from scratch, as shown in Table 4(a).

**Extension to other backbones**  As demonstrated in Figure 8(b), we choose Transolver as the default backbone of GeoPT, whose performance is also verified in recent research (Zhou et al., 2026). It is worth noticing that GeoPT is an architecture-agnostic pre-training model. Therefore, it is also possible to apply lifted geometric pre-training to GNOT (Hao et al., 2023) and Galerkin Transformer (Cao, 2021). As shown below, GeoPT can also bring improvements to these models.

*Table 4.* GeoPT Extension to time-dependent simulation and other backbones. All metrics are tested under full-data and full-training.

*(a) Time-dependent simulation of DTCHull.*

| RELATIVE L2 | TRANSIENT DTCHULL |
| --- | --- |
| FROM SCRATCH | 0.06189 |
| GEOPT | 0.05402 |

*(b) Applying GeoPT to other backbones.*

| RELATIVE L2 | DRIVAERML | NASA-CRM | AIRCRAFT | DTCHULL | CAR-CRASH |
| --- | --- | --- | --- | --- | --- |
| GALERKIN (2021) | 0.127 | 0.109 | 0.136 | 0.210 | 0.208 |
| GEOPT-GALERKIN | 0.083 | 0.095 | 0.099 | 0.156 | 0.182 |
| GNOT (2023) | 0.152 | 0.105 | 0.122 | 0.211 | 0.400 |
| GEOPT-GNOT | 0.116 | 0.097 | 0.099 | 0.180 | 0.219 |

**Align with existing benchmarks**  In this paper, we fix the training samples around 100 cases to mimic the industrial design practice. Here, we further conduct new experiments under aligned training settings with the two latest works: GAOT (Wen et al., 2025) and Transolver++ (Luo et al., 2025), to further benchmark GeoPT. As shown in Table 5, in the two largest datasets, DrivAerML (surface-only) and NASA-CRM, both Transolver and GeoPT surpass the previous benchmark by a large margin. As for AirCraft, GeoPT can also advance Transolver to the state-of-the-art performance. These results further justify the effectiveness of GeoPT, which consistently improves the model performance and achieves state-of-the-art.

---

[4]https://huggingface.co/tencent/Hunyuan3D-2/tree/main/hunyuan3d-vae-v2-0-withencoder

*Table 5.* Comparison between GeoPT and the previous method under aligned training settings. The results directly adopted from the previous paper are marked as ∗, where the results in DrivAerML (surface-only) and NASA-CRM are both from GAOT (2025) and AirCraft is from Transolver++ (2025). † marks our implementation. (a-b) The MSE and Mean AE are calculated on the normalized values.

*(a)* DrivAerML surface (450 training samples)

| PRESSURE COE | MSE | MEAN AE |
|---|---|---|
| GAOT∗ (2025) | 0.051729 | 0.12352 |
| GINO∗ (2023B) | 0.088124 | 0.15238 |
| TRANSOLVER∗ | *does not compare* | |
| TRANSOLVER† | 0.004223 | 0.04125 |
| GEOPT† | **0.003370** | **0.03617** |

*(b)* NASA-CRM (105 training samples)

| PRESSURE | MSE | MEAN AE |
|---|---|---|
| GAOT∗ (2025) | 0.077170 | 0.16014 |
| GINO∗ (2023B) | 0.105688 | 0.17450 |
| TRANSOLVER∗ | *does not compare* | |
| TRANSOLVER† | 0.011246 | 0.04938 |
| GEOPT† | **0.010722** | **0.04456** |

*(c)* AirCraft (140 training samples)

| AERODYNAMIC COES | RELATIVE L2 |
|---|---|
| TRANSOLVER++∗ (2025) | 0.064 |
| GINO∗ (2023B) | 0.133 |
| TRANSOLVER∗ | 0.092 |
| TRANSOLVER† | 0.083 |
| GEOPT† | **0.062** |

**Scaling performance**   To supplement the scaling results in Fig. 6 of the main text, we further test the corresponding performance in DTCHull and Car-Crash in Table 6. It is also observed that pre-training with GeoPT can help avoid potential overfitting, especially in industrial design tasks where only limited data is available. Besides, GeoPT can also take advantage of diverse geometry and dynamics conditions, highlighting the value of rich 3D geometry assets.

*Table 6.* Quantitative results for scaling performance and supplement results in DTCHull and Car-Crash. Here "w/o GeoPT" refers to Transolver trained from scratch. "Unique Geo" and "Dyn" represent the use ratio of geometry data and sampled dynamics, respectively.

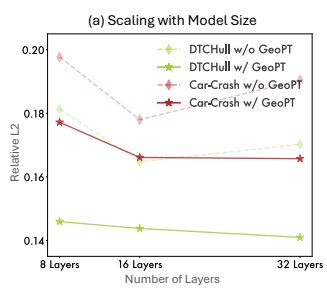

(a) Scaling with Model Size

| MODEL | LAYERS | DRIVAERML | NASA-CRM | AIRCRAFT | DTCHULL | CAR-CRASH |
|---|---|---|---|---|---|---|
| W/O GEOPT | 8 | 0.0927 | 0.0996 | 0.1104 | 0.1814 | 0.1977 |
| W/O GEOPT | 16 | 0.0847 | 0.0966 | 0.1160 | 0.1648 | 0.1779 |
| W/O GEOPT | 32 | 0.0839 | 0.0989 | 0.1153 | 0.1703 | 0.1905 |
| GEOPT | 8 | 0.0746 | 0.0880 | 0.0904 | 0.1459 | 0.1772 |
| GEOPT | 16 | 0.0702 | 0.0868 | 0.0864 | 0.1438 | 0.1661 |
| GEOPT | 32 | 0.0687 | 0.0851 | 0.0836 | 0.1410 | 0.1657 |

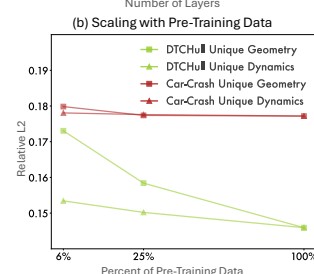

(b) Scaling with Pre-Training Data

| MODEL | RATIO | DRIVAERML | NASA-CRM | AIRCRAFT | DTCHULL | CAR-CRASH |
|---|---|---|---|---|---|---|
| UNIQUE GEO | 6% | 0.0819 | 0.0969 | 0.1102 | 0.1731 | 0.1798 |
| UNIQUE GEO | 25% | 0.0760 | 0.0902 | 0.0996 | 0.1584 | 0.1774 |
| UNIQUE GEO | 100% | 0.0746 | 0.0880 | 0.0904 | 0.1459 | 0.1772 |
| UNIQUE DYN | 6% | 0.0760 | 0.0913 | 0.1097 | 0.1534 | 0.1781 |
| UNIQUE DYN | 25% | 0.0755 | 0.0891 | 0.0954 | 0.1502 | 0.1776 |
| UNIQUE DYN | 100% | 0.0746 | 0.0880 | 0.0904 | 0.1459 | 0.1772 |

**Statistical test**   In Table 7, we also include the standard deviation of key results under 5 runs for statistical analysis. All training-data-saving claims are supported by 95% confidence intervals, where the equivalence margin is set as 0.001.

*Table 7.* Standard deviation of relative L2 under 5 runs.

| Standard deviation ($\times 10^{-3}$) | DrivAerML | NASA-CRM | AirCraft | DTCHull | Car-Crash |
|---|---|---|---|---|---|
| From Scratch 100% Training data | 0.43 | 0.94 | 1.03 | 1.34 | 0.71 |
| GeoPT 100% Training data | 0.68 | 0.35 | 1.23 | 0.88 | 0.31 |
| GeoPT 80% Training data | 0.80 | 0.66 | 1.27 | 1.10 | 0.56 |
| GeoPT 60% Training data | 0.81 | 0.93 | 1.17 | 0.49 | 0.13 |
| GeoPT 40% Training data | 0.31 | 0.76 | 1.83 | 0.91 | 0.76 |

**Quantitative results**   Here, we present the concrete values in Tables 8-12 for five simulation tasks in Fig. 1 and 5. We color some results to highlight GeoPT's capability in improving performance, convergence and reducing data requirements:

*(i) Improving performance:* We color results that outperform Transolver under the same samples and epochs as `bright blue`.

*(ii) Accelerating convergence:* Under the same training samples, the results surpass the best performance achieved by Transolver trained from scratch, are marked in `blue`, which indicates the convergence speed.

*(iii) Reducing data requirements:* The results better than Transolver under full-samples-full-epochs are marked in `dark blue`.

*Table 8.* Experiments on DrivAerML (Ashton et al., 2024), where we gradually increase the training data from 20 samples to 100 samples and training epochs from 50 epochs to 200 epochs. The relative L2 of Transolver (a) trained from scratch, (b) pre-trained with vector distance supervision, (c) trained with geometry condition, and (d) pre-trained with GeoPT are recorded.

| DATA \ EPOCH | (a) Transolver 50 | 100 | 150 | 200 | (b) w/ Geometry Pre-training 50 | 100 | 150 | 200 | (c) w/ Geometry Condition 50 | 100 | 150 | 200 | (d) w/ GeoPT (Ours) 50 | 100 | 150 | 200 |
|---|---|---|---|---|---|---|---|---|---|---|---|---|---|---|---|---|
| 20 | 0.374 | 0.272 | 0.231 | 0.206 | 0.498 | 0.525 | 0.394 | 0.341 | 0.443 | 0.362 | 0.289 | 0.264 | 0.216 | 0.179 | 0.160 | 0.148 |
| 40 | 0.328 | 0.199 | 0.193 | 0.140 | 0.486 | 0.424 | 0.357 | 0.256 | 0.339 | 0.264 | 0.191 | 0.169 | 0.165 | 0.132 | 0.121 | 0.111 |
| 60 | 0.215 | 0.170 | 0.135 | 0.112 | 0.486 | 0.320 | 0.278 | 0.230 | 0.299 | 0.175 | 0.147 | 0.119 | 0.150 | 0.113 | 0.100 | 0.091 |
| 80 | 0.206 | 0.142 | 0.108 | 0.102 | 0.322 | 0.271 | 0.237 | 0.214 | 0.239 | 0.159 | 0.133 | 0.102 | 0.133 | 0.103 | 0.089 | 0.084 |
| 100 | 0.197 | 0.120 | 0.101 | 0.093 | 0.326 | 0.251 | 0.255 | 0.202 | 0.200 | 0.133 | 0.099 | 0.094 | 0.118 | 0.092 | 0.080 | 0.075 |

*Table 9.* Experiments on NASA-CRM (Bekemeyer et al., 2025). The relative L2 error under different training data and epochs is recorded.

| DATA \ EPOCH | (a) Transolver 50 | 100 | 150 | 200 | (b) w/ Geometry Pre-training 50 | 100 | 150 | 200 | (c) w/ Geometry Condition 50 | 100 | 150 | 200 | (d) w/ GeoPT (Ours) 50 | 100 | 150 | 200 |
|---|---|---|---|---|---|---|---|---|---|---|---|---|---|---|---|---|
| 21 | 0.516 | 0.412 | 0.351 | 0.300 | 0.435 | 0.350 | 0.342 | 0.290 | 0.458 | 0.390 | 0.347 | 0.321 | 0.289 | 0.268 | 0.259 | 0.250 |
| 42 | 0.437 | 0.299 | 0.225 | 0.190 | 0.397 | 0.277 | 0.248 | 0.194 | 0.394 | 0.299 | 0.271 | 0.230 | 0.226 | 0.187 | 0.169 | 0.160 |
| 63 | 0.329 | 0.241 | 0.168 | 0.137 | 0.286 | 0.210 | 0.178 | 0.163 | 0.343 | 0.253 | 0.208 | 0.188 | 0.173 | 0.127 | 0.123 | 0.114 |
| 84 | 0.282 | 0.170 | 0.123 | 0.115 | 0.276 | 0.157 | 0.155 | 0.116 | 0.319 | 0.223 | 0.175 | 0.139 | 0.142 | 0.108 | 0.099 | 0.094 |
| 105 | 0.273 | 0.150 | 0.112 | 0.100 | 0.237 | 0.162 | 0.121 | 0.099 | 0.277 | 0.193 | 0.167 | 0.112 | 0.124 | 0.099 | 0.093 | 0.088 |

*Table 10.* Experiments on AirCraft (Luo et al., 2025). The relative L2 error under different training data and epochs is recorded.

| DATA \ EPOCH | (a) Transolver 50 | 100 | 150 | 200 | (b) w/ Geometry Pre-training 50 | 100 | 150 | 200 | (c) w/ Geometry Condition 50 | 100 | 150 | 200 | (d) w/ GeoPT (Ours) 50 | 100 | 150 | 200 |
|---|---|---|---|---|---|---|---|---|---|---|---|---|---|---|---|---|
| 20 | 0.586 | 0.401 | 0.207 | 0.172 | 0.595 | 0.486 | 0.365 | 0.247 | 0.567 | 0.266 | 0.211 | 0.186 | 0.567 | 0.264 | 0.197 | 0.170 |
| 40 | 0.319 | 0.198 | 0.195 | 0.138 | 0.562 | 0.321 | 0.296 | 0.237 | 0.317 | 0.218 | 0.191 | 0.142 | 0.240 | 0.159 | 0.147 | 0.122 |
| 60 | 0.226 | 0.247 | 0.138 | 0.122 | 0.519 | 0.304 | 0.317 | 0.213 | 0.299 | 0.234 | 0.163 | 0.124 | 0.224 | 0.159 | 0.121 | 0.107 |
| 80 | 0.197 | 0.175 | 0.158 | 0.138 | 0.360 | 0.199 | 0.189 | 0.191 | 0.259 | 0.214 | 0.153 | 0.125 | 0.151 | 0.135 | 0.113 | 0.103 |
| 100 | 0.153 | 0.136 | 0.137 | 0.110 | 0.199 | 0.165 | 0.170 | 0.199 | 0.238 | 0.195 | 0.161 | 0.118 | 0.134 | 0.117 | 0.101 | 0.090 |

*Table 11.* Experiments on DTCHull. The relative L2 error under different training data and epochs is recorded.

| DATA \ EPOCH | (a) Transolver 50 | 100 | 150 | 200 | (b) w/ Geometry Pre-training 50 | 100 | 150 | 200 | (c) w/ Geometry Condition 50 | 100 | 150 | 200 | (d) w/ GeoPT (Ours) 50 | 100 | 150 | 200 |
|---|---|---|---|---|---|---|---|---|---|---|---|---|---|---|---|---|
| 20 | 0.525 | 0.444 | 0.404 | 0.388 | 0.504 | 0.464 | 0.373 | 0.426 | 0.433 | 0.435 | 0.426 | 0.404 | 0.294 | 0.258 | 0.229 | 0.229 |
| 40 | 0.381 | 0.303 | 0.228 | 0.230 | 0.411 | 0.349 | 0.242 | 0.276 | 0.359 | 0.359 | 0.337 | 0.330 | 0.196 | 0.182 | 0.177 | 0.180 |
| 60 | 0.343 | 0.226 | 0.209 | 0.190 | 0.339 | 0.333 | 0.211 | 0.223 | 0.325 | 0.294 | 0.304 | 0.271 | 0.164 | 0.158 | 0.163 | 0.160 |
| 80 | 0.286 | 0.210 | 0.198 | 0.186 | 0.330 | 0.269 | 0.231 | 0.200 | 0.303 | 0.278 | 0.250 | 0.242 | 0.159 | 0.154 | 0.152 | 0.155 |
| 100 | 0.240 | 0.211 | 0.198 | 0.181 | 0.304 | 0.219 | 0.225 | 0.190 | 0.284 | 0.254 | 0.247 | 0.211 | 0.149 | 0.149 | 0.146 | 0.146 |

*Table 12.* Experiments on Car-Crash. The relative L2 error under different training data and epochs is recorded.

| DATA \ EPOCH | (a) Transolver 50 | 100 | 150 | 200 | (b) w/ Geometry Pre-training 50 | 100 | 150 | 200 | (c) w/ Geometry Condition 50 | 100 | 150 | 200 | (d) w/ GeoPT (Ours) 50 | 100 | 150 | 200 |
|---|---|---|---|---|---|---|---|---|---|---|---|---|---|---|---|---|
| 20 | 0.490 | 0.440 | 0.384 | 0.364 | 0.386 | 0.332 | 0.306 | 0.294 | 0.474 | 0.421 | 0.391 | 0.365 | 0.356 | 0.311 | 0.278 | 0.260 |
| 40 | 0.438 | 0.358 | 0.318 | 0.282 | 0.398 | 0.354 | 0.306 | 0.296 | 0.421 | 0.361 | 0.328 | 0.298 | 0.296 | 0.263 | 0.232 | 0.219 |
| 60 | 0.389 | 0.319 | 0.274 | 0.217 | 0.370 | 0.307 | 0.280 | 0.273 | 0.395 | 0.330 | 0.293 | 0.256 | 0.268 | 0.237 | 0.209 | 0.194 |
| 80 | 0.357 | 0.288 | 0.231 | 0.202 | 0.338 | 0.301 | 0.269 | 0.244 | 0.368 | 0.315 | 0.255 | 0.233 | 0.252 | 0.211 | 0.194 | 0.185 |
| 100 | 0.337 | 0.254 | 0.205 | 0.198 | 0.318 | 0.268 | 0.248 | 0.225 | 0.345 | 0.274 | 0.238 | 0.193 | 0.236 | 0.198 | 0.184 | 0.177 |

**Representation visualization** Due to space limitations in the main text, we present the complete visualization of the physics states in Fig. 22-25 as a supplement to Fig. 3 and 7, which includes the learned correlations under various supervisions and dynamics-dependent correlations prompted by different configurations of dynamics conditions.

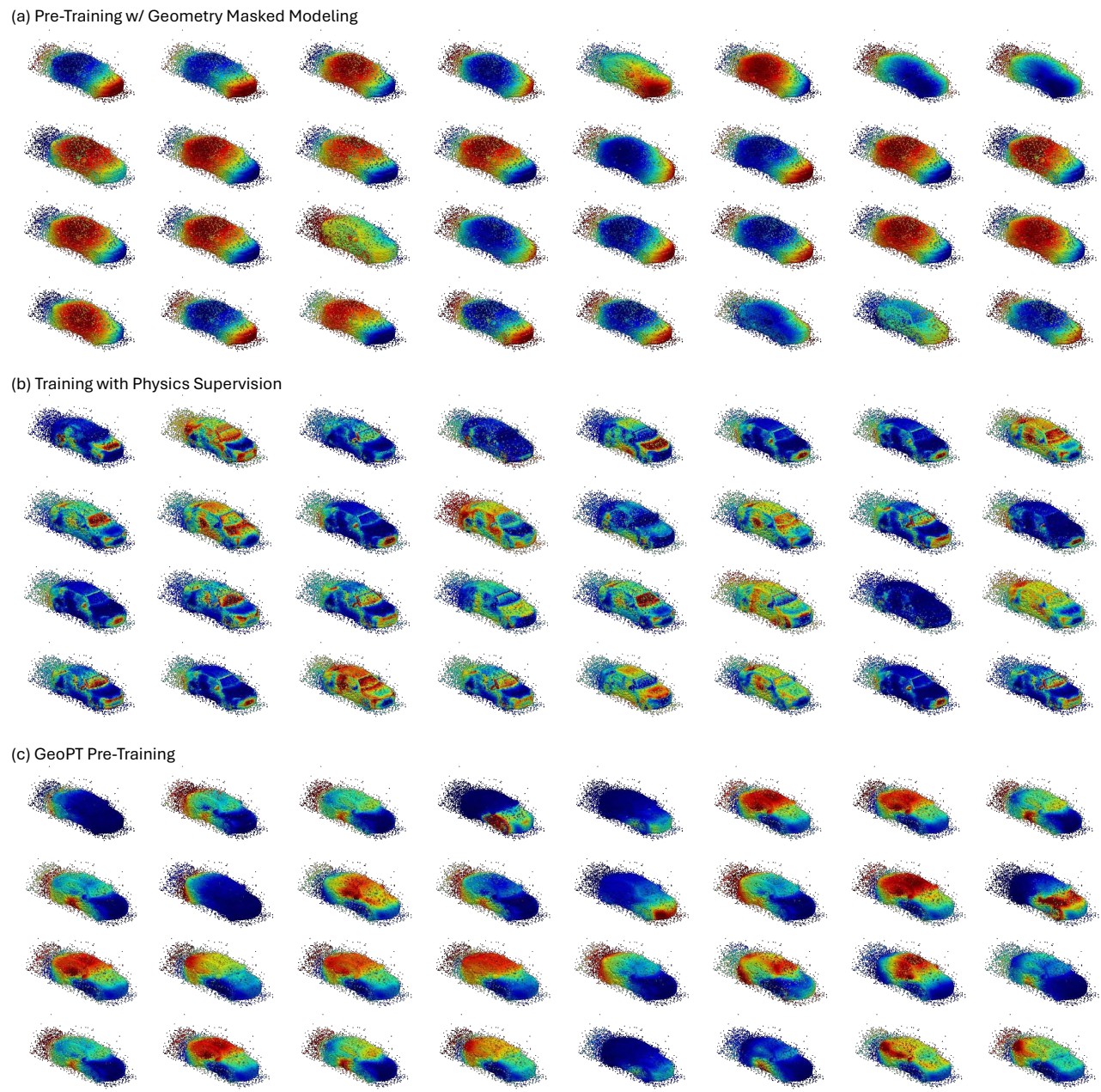

*Figure 22.* Visualization of physical states learned from different supervision signals, including (a) pre-training by learning to predict vector distance, (b) directly training with DrivAerML physics supervision and (c) pre-training with the dynamic process proposed by GeoPT. Here, we visualize the last layer. The visualizations from other layers can differ in absolute value but share a similar distribution.

(a) GeoPT with +x direction, 0.3 normalized speed

(b) GeoPT with 60° shifted direction, 0.3 normalized speed

(c) GeoPT with -60° shifted direction, 0.3 normalized speed

(d) GeoPT with +x direction, 0 normalized speed

(e) GeoPT with +x direction, 2 normalized speed

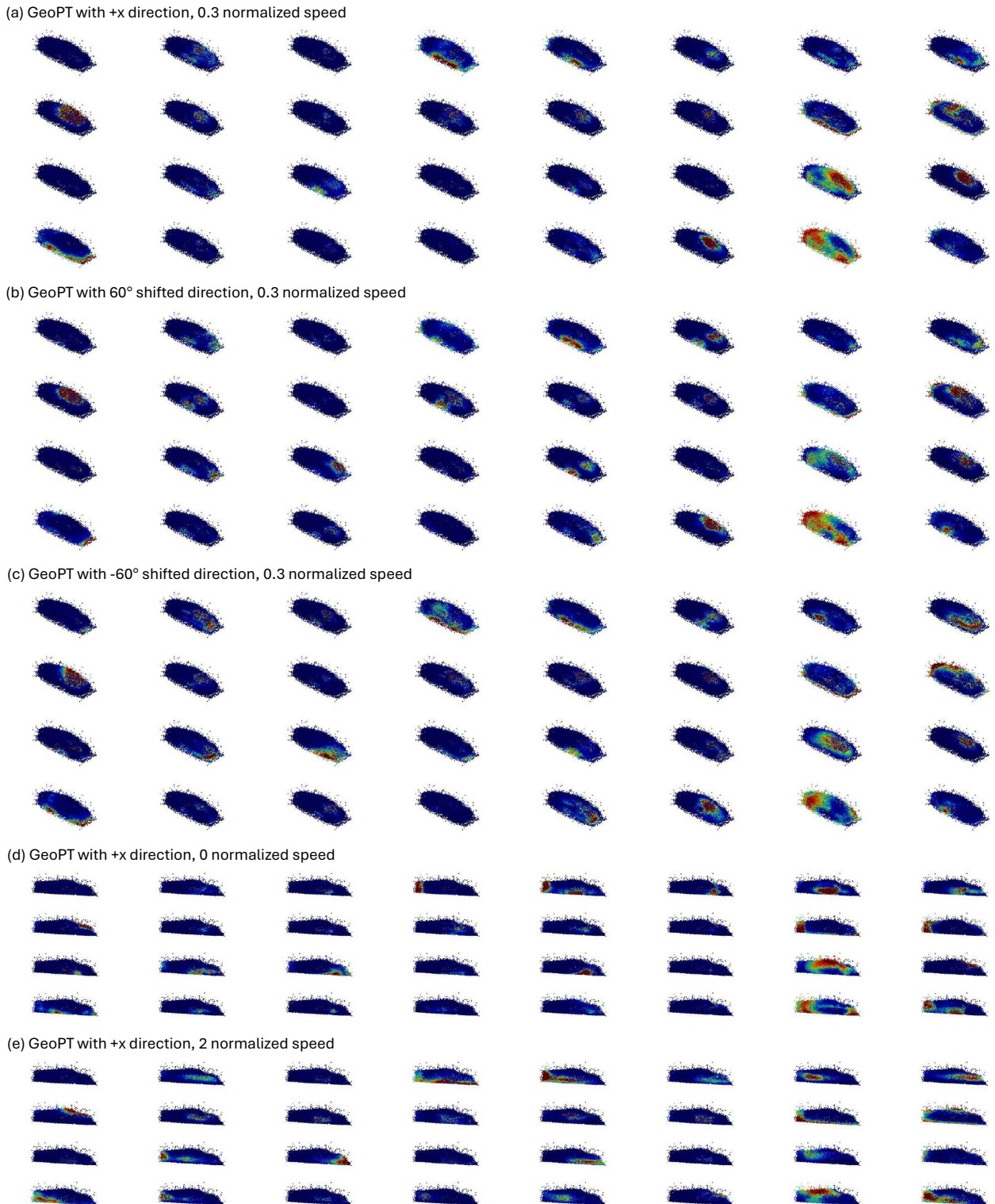

*Figure 23.* Visualization of DrivAerML physical states in GeoPT "prompted" from different dynamics configurations, including varied (a-c) direction and (d-e) velocity norm (speed) configurations. Here, we visualize the fourth layer as a supplement to the last layer visualization in Fig. 22. Notably, the states in the fourth layer are less distinguishable in the middle layers when compared with the last layer. This can be viewed as an architectural feature of Transolver, which can enable better global interaction among different states.

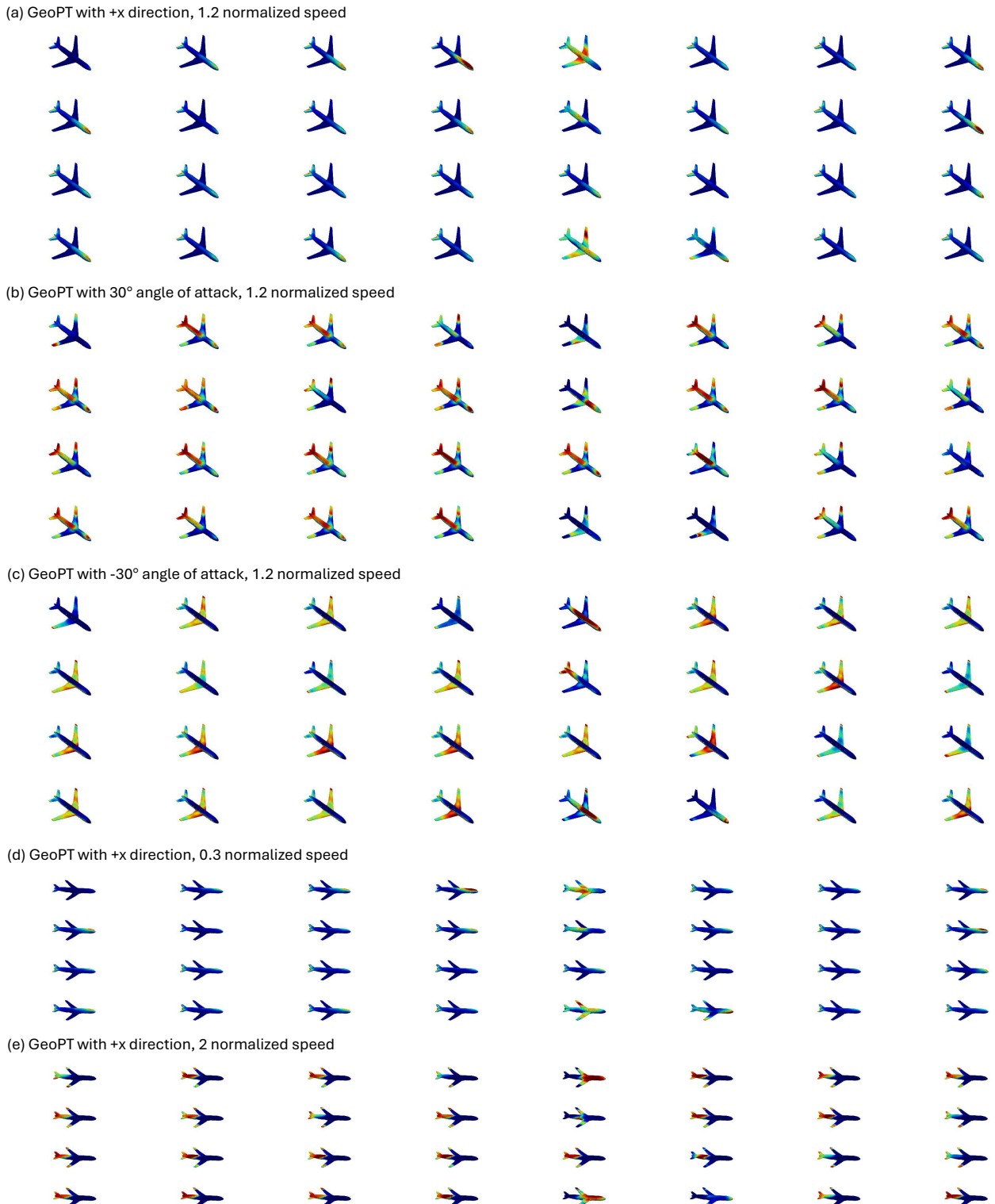

*Figure 24.* Visualization of NASA-CRM physical states in GeoPT "prompted" from different dynamics configurations, including varied (a-c) direction and (d-e) speed configurations. Here, we visualize the physical states in the last layer. Similar to DrivAerML, changing the angle of attack in the $y$-$z$ plane will affect the state distribution and increasing speed will lead to a more concentrated state distribution.

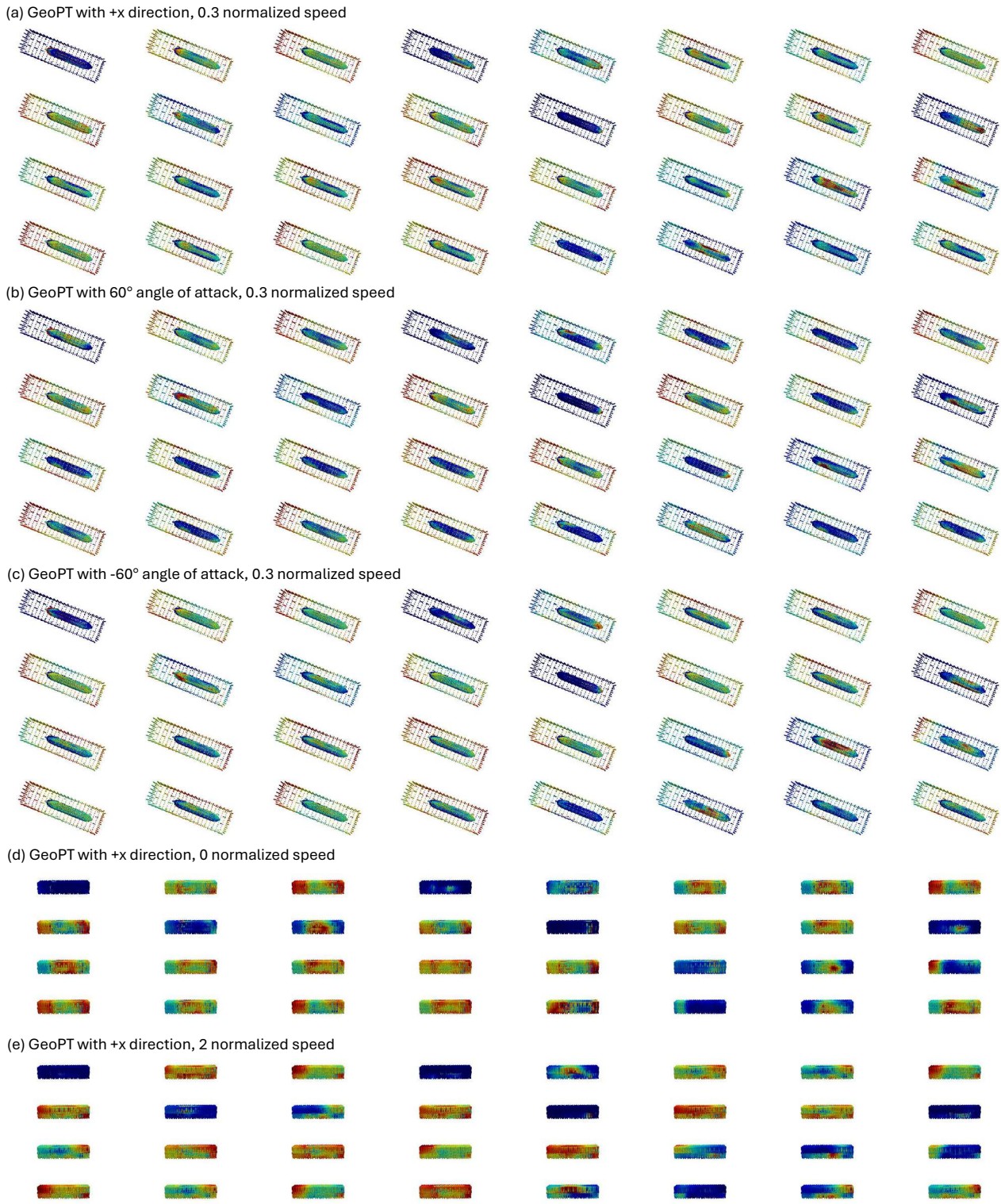

*Figure 25.* Visualization of DTCHull physical states in GeoPT "prompted" from different dynamics configurations, including varied (a-c) direction and (d-e) speed configurations. Here, we visualize the physical states in the last layer. Notably, to ensure a clear presentation of the inside ship surface, we downsample the mesh by 10 times for visualization.

# G. Limitations and Future Work

This paper provides a scalable pathway to utilize off-the-shelf geometries for pre-training neural simulators and has demonstrated effectiveness in extensive industrial-fidelity simulation tasks. Despite favorable performance and generalizability, GeoPT also has some limitations, lying in the following aspects.

In GeoPT, we propose to parameterize the diverse simulation settings into a point-wise dynamics field, which can cover the diversity in geometry, flow direction and speed, as well as material differences of typical simulation benchmarks. Although this is one step forward in unifying diverse simulation tasks but still cannot precisely describe all kinds of settings. For example, in the crash simulation that considers both elastic and strength properties of materials, the most reasonable to parameterize two properties into one dynamic speed value for every element, while this may lose some distinguishability due to the dimension reduction. However, since we only focus on the pre-training process, such a limitation can be resolved by extending the input channel and tuning new parameters with zero initialization, like ControlNet (Zhang et al., 2023). In the future, we would like to explore a more general framework to unify diverse simulation tasks.

GeoPT primarily targets simulations involving complex geometries, a common requirement in industrial design and a setting where neural simulators have achieved their most notable successes. Additionally, it is also interesting to see the extension of GeoPT to simulations without complex geometry boundaries, such as 3D turbulence in regular grids (Perlman et al., 2007). Since such simulations are mainly about computationally intensive physics interactions, the previous explorations (Holzschuh et al., 2026) still rely on expensive physics supervision. Taking such regular grid simulation into account, one possible way is to leave a certain ratio of pre-training iterations for dynamics under an empty geometry boundary during the pre-training process of GeoPT, where all the tracking points are in free-flight motion, which can make the model learn the intrinsic isotropic dynamics. We would like to leave this exploration as future work.

