# OpenReview forum: "GeoPT: Scaling Physics Simulation via Lifted Geometric Pre-Training"
_ICML.cc/2026/Conference — ICML 2026 regular_

### Official Review · Reviewer_ZcS3 · 2026-03-09

**Soundness:** 4
**Presentation:** 4
**Significance:** 4
**Originality:** 4
**Overall Recommendation:** 5
**Confidence:** 4

**Summary:**

This paper introduces GeoPT, a novel approach aimed at scaling neural simulation in physics through the Lifted Geometric Pre-Training. The main idea is to bridge the gap between geometry and physics by pre-training neural simulators using abundant geometric data, while incorporating synthetic dynamics. This approach enables the models to leverage geometry-dynamics coupled representations during pre-training without needing labeled physics data. GeoPT significantly improves simulation performance across a variety of industrial-scale benchmarks, including fluid mechanics (for cars, aircraft, and ships) and solid mechanics (for crash simulations). The model achieves 20-60% reduction in data requirements and accelerates convergence by up to 2x.

**Compliance With Llm Reviewing Policy:**

Affirmed.

**Final Justification:**

The rebuttal has adequately addressed my concerns. The new experiments on Galerkin Transformer and GNOT validate the architecture-agnostic claim; the 5-run statistics confirm the robustness of data-saving claims; and the transient DTCHull result extends applicability beyond steady-state. The new theorem formalizing native pre-training as a degenerate case of lifted pre-training further strengthens the theoretical grounding. I raise my score from 4 to 5 (Accept).

**Key Questions For Authors:**

1. The pre-training process involves different datasets, mesh sizes, and varying input/output feature dimensions. How is the training unified across these differences? Additionally, do the input/output feature dimensions in the downstream tasks remain consistent with those during pre-training? If not, how does the model adapt during the fine-tuning stage?

2. While it is claimed that the approach is architecture-agnostic, most experiments are conducted using the Transolver backbone. Could you show lifted pre-training benefits on at least one non-Transolver backbone (e.g., GNOT or Galerkin Transformer)? Also, how does GeoPT compare with recent physics foundation models in data efficiency and simulation accuracy?

3. Despite the large performance improvements, there is a lack of statistical analysis (e.g., error bars, confidence intervals, multiple runs). Could you provide more detailed statistical analysis on these aspects?

4. Have you explored alternative synthetic dynamics beyond velocity? This would clarify whether the specific lifting choice matters or any dynamics augmentation suffices.

**Limitations:**

yes

**Strengths And Weaknesses:**

**Strengths**
- The methodology is technically sound, underpinned by strong theoretical foundations, including the use of synthetic velocity fields for dynamics-aware pre-training. The experiments demonstrate consistent improvements in data efficiency, performance, and convergence speed across several complex simulation tasks.
- The paper is well-structured and clearly presents the core methodology and experimental results. The theoretical explanations are sound, with sufficient detail on the approach's innovations and how they address existing challenges. Visual aids, such as figures and tables, effectively complement the text, offering clarity on experimental setups and results.
- This work makes a certain contribution to the field of neural simulators in physics, particularly in improving the scalability of pre-training models. The ability to scale simulation without the need for expensive labeled physics data addresses a critical bottleneck in industries like automotive and aerospace, which rely heavily on simulation tasks.
- The dynamics-lifted geometric pre-training is novel and provides a fresh perspective for scaling neural simulators. The paper advances the field by presenting a new way to utilize geometry data and introduces synthetic dynamics to improve downstream physics tasks. This approach stands apart from existing work in the field by eliminating the need for physics-labeled data during pre-training.

**Weaknesses**
- While the approach is innovative, the paper does not provide a deep theoretical justification for the choice of dynamics-lifted supervision over other potential unsupervised learning paradigms. Additionally, no error bars or confidence intervals are reported across any experiment. Given small test sets (20–50 samples) and stochastic training, claims like "60% data saving" may be sensitive to random seeds.
- The fine-tuning configuration for the synthetic velocity norm ($V_S$) relies on a somewhat heuristic presentation. While the authors provide a general "recipe" for low and high-speed simulations, the presentation lacks a rigorous, systematic method or sensitivity analysis for mapping real-world physical dimensions (like Reynolds or Mach numbers) to these normalized velocity magnitudes.
- The practical impact is somewhat limited by the need for manual velocity field design per task, which reduces plug-and-play appeal for new physics domains. Additionally, all benchmarks focus on steady-state or time-aggregated quantities, while transient simulations are equally important in industrial practice. Finally, all experiments use only the Transolver backbone, leaving the claimed architecture-agnostic generality unvalidated.
- The synthetic dynamics choice (uniform velocity + sticking boundary) is simple and unexplored in alternatives. It remains unclear whether this specific choice is critical or whether other lifting strategies could work equally well or better.

---

> ### Author Rebuttal · Authors · 2026-03-29
>
> Many thanks to Reviewer ZcS3 for the detailed review and insightful questions.
>
> > **W1.1:** While approach is innovative, no theoretical justification for dynamics-lifted supervision over others.
>
> In the original submission, we have included:
>
> - Representation visualization in $\underline{\text{Figure 3(a)}}$. "Without dynamics, native pre-training cannot capture the geometry-dynamics coupling."
> - Theoretical interpretation in $\underline{\text{Remark 4.1}}$. Our supervision is equivalent to a conservation constraint.
>
> Here, we further established a new theorem.
>
> > **New Theorem:** Given optimal models $F_{\theta_{\text{native}}}^\ast,F_{\theta_{\text{lifted}}}^\ast$ achieving zero native space and lifted losses, then $\text{slice}(E_{V}[F_{\theta_{\text{native}}}^\ast(x;G,V)])=F_{\theta_{\text{lifted}}}^\ast(x;G)$, where $E$ denotes expectation, $\text{slice}$ is taking $t=0$.
>
> This can be proven based on topology structure and Eq.(5) on page 4. This proves that **native space pre-training learns the expectation of our lifted pre-training**, whose representation will degenerate to reduced space ($\underline{\text{Lines 75-77 of Introduction}}$).
>
> > **W1.2 & Q3:** Despite large improvements, no error bars.
>
> Since experiments are based on computation-heavy simulations, both train and test data are limited. Our setting (20-50 test samples) follows prior work: Transolver++ and GAOT.
>
> Here, we provide standard deviations for key results. All data-saving claims are supported by 95% confidence intervals under 0.001 equivalence margin.
>
> |5 runs Std (x$10^{-3}$)|DrivAerML|NASA|AirCraft|DTCHull|Crash|
> |-|-|-|-|-|-|
> |Transolver-100%data|0.43|0.94|1.03|1.34|0.71|
> |GeoPT-100%data|0.68|0.35|1.23|0.88|0.31|
> |GeoPT-80%data|0.80|0.66|1.27|1.10|0.56|
> |GeoPT-60%data|0.81|0.93|1.17|0.49|0.13|
> |GeoPT-40%data|0.31|0.76|1.83|0.91|0.76|
>
> > **W2 & W3.1:** While provides a general recipe, lacks sensitivity analysis for mapping real-world physical dimensions to normalized magnitudes. Need velocity design per task, somewhat limits practical impact.
>
> We have provided sensitivity analyses in $\underline{\text{Figures 12-13}}$, where a distinguishable configuration is sufficient for GeoPT in most cases ($\underline{\text{Remark D.1}}$).
>
> As for mapping real-world physical dimensions, we admit this is an open problem in physics foundation models and none of the previous work presented a solution. GeoPT's recipe can be an initial step and empirical exploration. We will discuss this in future work and limitations.
>
> Notably, as a general method, GeoPT usage is much easier than numerical solvers that need super complex scripts per task. And GeoPT only needs to configure one additional parameter (velocity magnitude) than conventional neural surrogates.
>
> > **W3.2:** No transient simulations.
>
> We applied GeoPT to transient simulation DTCHull, which is pre-time-aggregated DTCHull and requires simulating the next transient field based on the current. Unlike steady-state, transient field correlation is more short-term. Thus, we configure the velocity norm 5x smaller than steady-state for fine-tuning, where GeoPT still brings benefits.
>
> |Relative L2|Transient DTCHull|
> |-|-|
> |From Scratch|0.06189|
> |GeoPT|0.05402|
>
> > **W3.3 & Q2:** Show lifted pre-training benefits on other backbones.
>
> We applied GeoPT to Galerkin Transformer and GNOT, which brings consistent promotion.
>
> |Relative L2|DrivAerML|NASA|AirCraft|DTCHull|Crash|
> |-|-|-|-|-|-|
> |Galerkin-Scratch|0.127|0.109|0.136|0.210|0.208|
> |Galerkin-GeoPT|0.083|0.095|0.099|0.156|0.182|
> |GNOT-Scratch|0.152|0.105|0.122|0.211|0.400|
> |GNOT-GeoPT|0.116|0.097|0.099|0.180|0.219|
>
> > **W4 & Q4:** Synthetic dynamics is simple and unexplored in alternatives.
>
> As a pre-training method, we prefer to keep supervision as simple as possible to maintain generation efficiency and downstream generalization. And our current design is suitable (0.2s per sample generation & conservation law).
>
> See **Reviewer jPdy Q2** for alternative choice discussion, including divergence-free and boundary layer effects.
>
> > **Q1:** How to unify pre-training, adapt to fine-tuning?
>
> Pre-training: we sample the same number of tracking particles for all geometries (Table 3 in Appendix). The input includes geometry info (pos, sdf, normal) and dynamics info (direction and magnitude), and the output is 3-step vector distance, which is shared in all samples, thereby naturally aligned.
>
> Fine-tuning: the input dim remains the same, but the output dim varies according to required quantities. We replace the final linear projection for adaptation. We will include these details in the revision.
>
> > **Q2:** Compare with recent physics foundation models in data efficiency and simulation accuracy
>
> As discussed in related work, all the previous physics foundation models rely on simulation data for pre-training (thereby very slow in generating pre-training data) and none of them can handle industrial simulations with irregular geometries evaluated in this paper.

---

> > ### Author Rebuttal · Reviewer_ZcS3 · 2026-04-02
> >
> > Thank you for the detailed rebuttal, which has adequately addressed my concerns. This is an interesting and highly promising work with significant potential for further development. I have raised my score to 5.

---

> > > ### Author Response · Authors · 2026-04-02
> > >
> > > Dear Reviewer ZcS3,
> > >
> > > Thank you very much for recognizing our rebuttal and for raising your score. We also sincerely appreciate your thoughtful and detailed review, especially your suggestions on theoretical justification, statistical analysis, and backbone generalization. We will incorporate all the newly added experiments and theoretical clarifications into the revised paper.
> > >
> > > Thank you again for your time and dedication.

---

### Official Review · Reviewer_jPdy · 2026-03-11

**Soundness:** 3
**Presentation:** 3
**Significance:** 3
**Originality:** 3
**Overall Recommendation:** 5
**Confidence:** 3

**Summary:**

GeoPT addresses the high cost of high-fidelity physics simulations by leveraging 3D representations and introducing Lifted Geometric Pre-Training. This concept describes pre-training on geometries with solver-free synthetic physical trajectories. The model can learn physical meaning across different geometries, which helps in the Second step of fine-tuning to converge faster and offer better predictions across Fluid and Solid mechanics. The experimental result show that GeoPT achieves up to 60% better data efficiency and converges twice as fast as models trained from scratch.

**Compliance With Llm Reviewing Policy:**

Affirmed.

**Final Justification:**

The paper presents a well-formulated method to enhance physics simulations by reducing the amount of high-fidelity samples. The experiments underline the method and the rebuttal has addressed my open questions. Therefore, I support the acceptance of the paper.

**Key Questions For Authors:**

(Q1) How does the model perform on geometries out of distribution from ShapeNet? Have you considered looking at parts where you did not finetune on or samples that are completely out of the training range to see how well the model performs ?

(Q2) When looking at the synthetic velocity fields, how could one account for physical constraints like Divergence-Free flow or Boundary Layer effects? And do you think that by adding physical constraints one could improve the results?

(Q3) How does the performance of GeoPT scale as the complexity of the mesh (number of vertices/elements) increases?

**Limitations:**

Yes

**Strengths And Weaknesses:**

**Presentation**

The paper is well structured and provides a clear presentation of the problem and the introduced method. The reader is able to grasp the main ideas without indepth knowledge. How the experiments and the architectural choices are build up, makes a lot of sense and compliments the structure of the paper.

**Soundness**

The paper is technically sound and the choosen experiments undermine the chosen architectural and strategic decisions. Additionally, strong empirical evidence is seen in the experiments where fluid and solid problems were tested.

**Significance / Originality**

The paper shows potential to be applied in real-world industrial simulation. Not only does the paper demonstrate that less data is required, but it also that already generated 3D structures can be used as a basis for the model training.
The proposed approach further demonstrates a potential direction, how to solve the problem with high-fidelity data, instead of making simulations faster, the authors created a task that teaches the model to learn physical properties.

---

> ### Author Rebuttal · Authors · 2026-03-29
>
> We would like to sincerely thank Reviewer jPdy for providing valuable feedback and questions.
>
> > **Q1:** Model performance on geometries OOD from ShapeNet. Looking at parts where you did not finetune on or samples that are completely out of the training range.
>
> In our original submission, we have experimented with geometries out of distribution (OOD) from ShapeNet, including
>
> - AirCraft ($\underline{\text{Figure 5(b) of main text}}$): As visualized in $\underline{\text{Figures 5,16}}$, this dataset contains simulations for highly specialized aircraft with clean and smooth surfaces. And we have manually confirmed that the airplane subset of ShapeNet does not contain very similar geometries.
> - Radiosity ($\underline{\text{Last experiment of main text, Appendix A}}$): As visualized in $\underline{\text{Figures 1,9}}$, this dataset involves simulations for various bunny shapes, which are completely different from the pre-trained geometries (i.e., car, ship, airplane), resulting in an extreme OOD scenario.
>
> GeoPT consistently boosts performance in both AirCraft (40% reduction in training data requirements) and Radiosity (better high-frequency shadow modeling), thereby verifying its effectiveness.
>
> Besides, we also want to highlight that GeoPT offers a pathway to scale physics simulations with off-the-shelf geometries. And if we could include as diverse geometries as possible, the geometry OOD issue may not exist in the end, which is also listed as our future work in $\underline{\text{Sec. 6 of main text}}$.
>
> > **Q2:** When looking at the synthetic velocity fields, how could one account for physical constraints like Divergence-Free flow or Boundary Layer effects? By adding physical constraints, one could improve the results.
>
> It is a brilliant idea to incorporate new physical constraints during pre-training. Especially, the divergence-free constraint could be helpful for incompressible fluid, and boundary layer effects are essential in simulating viscous flow.
>
> However, since GeoPT attempts to construct a pre-trained model for general physics, these two constraints may not be as general as mass conservation that we currently use. Here are detailed analyses.
>
> **(1) Regarding "divergence-free".**
>
> To transform the synthetic velocity of GeoPT into a divergence-free field, the classical method is to solve an additional Poisson equation to fix the velocity, which will cause additional computation costs, limiting the scalability of self-supervision data.
>
> Specifically, as listed below, simulating a divergence-free field will cause **27x running time** for self-supervision data generation than our official setting.
>
> |Supervision|Running Time per sample (80 CPU cores)|
> |-|-|
> |Mass conservation (official setting)|~0.2s|
> |Divergence-free constraint (fixed by solving Poisson equation)|~5.4s|
>
> Also, we want to highlight that there are some simulations, such as compressible flow, that do not satisfy the divergence-free constraint, but mass conservation still exists.
>
> **(2) Regarding "boundary layer effects".**
>
> Considering boundary layer effects will make the pre-training data align better with the downstream aerodynamics simulation and boost the performance in DrivAerML, NASA, AirCraft and DTCHull.
>
> However, the **Car-Crash and Radiosity** do not present special boundary layer effects. Incorporating additional constraints may damage performance in these tasks. Moreover, it will introduce additional hyperparameters (e.g., boundary layer splitting plane) to control the pre-training data generation and configure the fine-tuning dynamics prompt, limiting the practical usage.
>
> In conclusion, incorporating additional physical constraints can help certain relative simulations but may damage the model's generalizability or cause additional computation cost; therefore, it may not be a good choice in building a "foundation model".
>
> We would like to leave the exploration of additional physical constraints as our future work for domain-specific simulations.
>
> > **Q3:** How does the performance of GeoPT scale as the complexity of the mesh (number of vertices/elements) increases?
>
> As listed in $\underline{\text{Table 1 of main text}}$, experiments of GeoPT involve extremely high-resolution simulations (>100M elements per sample), which are the largest public benchmark to our best knowledge. Therefore, our experiments are sufficient to examine the model performance when scaling mesh complexity.
>
> Regarding the scaling curve, Transolver (the official backbone of GeoPT) has been proven to be resolution general in $\underline{\text{Figure 8(a) of their paper}}$, which performs relatively stable under various resolutions. As per your request, we further tested GeoPT under different resolutions, where the performance is also stable. Specifically, the model performs slightly better in a smaller mesh due to the simpler physics interaction.
>
> |DrivAerML|160M (Our official experiment)|1M|100K|
> |-|-|-|-|
> |GeoPT Relative L2|0.07459|0.07384|0.07372|

---

> > ### Author Rebuttal · Reviewer_jPdy · 2026-04-02
> >
> > Thank you for clarifying the open points. This submission represents a good contribution to the field.

---

> > > ### Author Response · Authors · 2026-04-02
> > >
> > > Dear Reviewer jPdy,
> > >
> > > Thank you for your thoughtful review and for acknowledging our rebuttal and contribution. We especially appreciate your insightful comments on OOD generalization, scale generalization, and alternative physical constraints. We will incorporate these points into the revised version. Thank you again for your time and dedication.

---

### Official Review · Reviewer_6Wws · 2026-03-13

**Soundness:** 2
**Presentation:** 3
**Significance:** 2
**Originality:** 3
**Overall Recommendation:** 4
**Confidence:** 3

**Summary:**

The paper introduces GeoPT, a novel framework designed to alleviate the data-bottleneck associated with high-fidelity physical simulations. The core innovation is "lifted geometric pre-training," which bridges the gap between static geometric data and dynamic physical simulations. By injecting synthetic dynamics into large-scale, off-the-shelf geometric datasets, the authors enable dynamics-aware self-supervised learning without the need for expensive ground-truth physical labels.

**Compliance With Llm Reviewing Policy:**

Affirmed.

**Final Justification:**

I thank the authors for providing clarifications to my concerns.

**Key Questions For Authors:**

- Generalization to Aerodynamic Structures:
 Can the authors provide and analyze results for the Airplane category (e.g., from ShapeNet) in addition to automotive scenes? While many vehicle shapes may exhibit somewhat similar vector field patterns across the dataset, aircraft possess more complex, streamlined geometries where the interplay between wings, fuselage, and tailfins is critical to fluid dynamics. How does GeoPT perform on these more sensitive aerodynamic structures compared to standard car shapes?

**Limitations:**

see above weaknesses

**Strengths And Weaknesses:**

**Strengths**
- Novelty in Problem Definition: The paper identifies a critical and well-defined bottleneck in the field: the significant gap between the abundance of "offline geometric data" and the scarcity of "high-fidelity physical simulation data." The primary contribution lies in the insight that geometry alone is insufficient for neural simulators. By proposing a new pre-training space—"Geometry + Synthetic Dynamics"—the authors successfully bridge this gap, providing a scalable pathway to pre-train models that are inherently aware of motion and physical change.

**Weaknesses**
- Limited Algorithmic Novelty: While the problem setting and the conceptual framework ("lifting") are highly novel, the specific learning objectives and architectural components do not introduce significant breakthroughs in machine learning methodology. The technical implementation relies largely on existing self-supervised learning paradigms, which may limit the paper’s contribution to core algorithmic theory.

- Risk of Underfitting and Averaging: Due to the vast diversity of geometric shapes and the generalized nature of synthetic dynamics during pre-training, the model may suffer from underfitting regarding specific physical laws or complex boundary conditions. Specifically, for objects with highly sensitive aerodynamic features (e.g., car curvatures), the model might output "averaged" dynamics. It may fail to capture subtle physical nuances—such as turbulence or pressure shifts—that vary critically with specific velocities, angles, or boundary layers.

---

> ### Author Rebuttal · Authors · 2026-03-29
>
> Many thanks to Reviewer 6Wws for providing the insightful review and questions.
>
> > **Clarify contribution**
>
> To avoid potential misunderstandings, we first summarize our contribution below. Notably, **the lifting framework belongs to our algorithmic design, instead of problem definition.**
>
> ||Problem Definition|Algorithmic Design|
> |-|-|-|
> |High-level|Scarce simulations, abundance geometries|Lifting pre-training|
> |Detailed|Geometry alone fails|Dynamics-lifted supervision|
>
> > **W1:** Limited Algorithmic Novelty: While setting and lifting framework are highly novel, learning objectives and architectural components do not introduce significant breakthroughs.
>
> Thanks for acknowledging our setting and lifting framework as "highly novel".
>
> **(1) "Learning objective" in GeoPT is distinct from prior, which is driven by our lifting framework and supported by conservation theory.**
>
> Our learning objective is the synthetic dynamics trajectory, which is the first dynamics-lifted supervision and is distinct from existing methods as discussed in related work. Besides, this learning objective has also been proven to be a conservation constraint ($\underline{\text{Remark 4.1}}$).
>
> Notably, the learning objective is the core of GeoPT. Without this, we cannot implement and validate our lifting idea. Thus, **the lifting framework and learned objectives should be considered as integral parts**.
>
> Besides, we prove a `New Theorem (see Review ZcS3 W1.1)`, which shows existing self-supervision paradigms essentially learns the expectation of our design, highlighting our algorithmic theory contribution.
>
> **(2) "Architectural components" are not the focus of architecture-agnostic GeoPT.**
>
> Architecture-agnostic is a special feature of GeoPT (main text Line275-right), rather than a technical simplification. To demonstrate this, we further apply GeoPT to a new Transformer solver, and GeoPT consistently improves it.
>
> See `Review ZcS3 W3.3 & Q2` for more results.
>
> |Relative L2-Galerkin Transformer|DrivAerML|NASA-CRM|AirCraft|DTCHull|Crash|
> |-|-|-|-|-|-|
> |From Scratch|0.127|0.109|0.136|0.210|0.208|
> |GeoPT|0.083|0.095|0.099|0.156|0.182|
>
> Notably, GeoPT aims to construct a scalable framework. As commonly acknowledged by foundation model research, we prefer a neat and general framework over complicated designs.
>
> > **W2:** Risk of Underfitting and Averaging: Due to data diversity, the model may suffer from underfitting. For sensitive aerodynamics (e.g., car curvatures), model might output averaged dynamics.
>
> Thanks for this insightful question.
>
> **(1)  Sensitive aerodynamic experiments: Averaging does not happen.**
>
> First, GeoPT follows the pre-training fine-tuning paradigm, where the fine-tuning can help to capture subtle physics. Thus, even if model capacity is limited for pre-training data, it will not output averaged dynamics.
>
> As shown in Figure 5, GeoPT improves the SOTA backbone in five industrial-quality benchmarks, where DrivAerML and NASA-CRM exactly match your sensitive aerodynamics setting (cases slightly different in geometry and include turbulence; see dataset papers for visualization). GeoPT can precisely simulate the wake flow of cars and wing flow of airplanes (showcases in $\underline{\text{Figures 7,15}}$).
>
> Besides, as shown below, GeoPT is much better than the averaged physics fields, demonstrating that trivial averaging does not happen.
>
> |Relative L2|DrivAerML|NASA-CRM|
> |-|-|-|
> |Test Set Average|0.7380|0.5798|
> |GeoPT|0.0746|0.0880|
>
> **(2) Model scaling experiments: underfitting can be resolved by increasing model size.**
>
> We agree with the reviewer that our generated pre-training data is highly diverse. However, we would like to view this as the promising potential of GeoPT, rather than a weakness, since it means our pre-training data contains sufficient physical knowledge to be learned.
>
> Actually, just because of data diversity, GeoPT presents favorable model scaling properties. As visualized in $\underline{\text{Figures 6-7,15-18}}$, a larger model can capture subtle physics better.
>
> Furthermore, we newly pre-train GeoPT on the whole ShapeNet (51,300 geometries, 5x larger). Under a limited model size, the performance gain faces saturation, but will not decrease due to the fine-tuning. And such underfitting saturation can be broken by increasing model size (32-layer results).
>
> |Relative L2|DrivAerML|NASA-CRM|
> |-|-|-|
> |8layers-3 subsets|0.0746|0.0880|
> |8layers-whole ShapeNet|0.0746|0.0876|
> |32layers-3 subsets|0.0687|0.0851|
> |32layers-whole ShapeNet|0.0679|0.0803|
>
> > **Q1:** Generalization to Aerodynamics: provide and analyze Airplane results.
>
> We have provided airplane results in the original submission, including quantitative results, error map and representation visualizations:
>
> - NASA-CRM (NASA 3D airplane): $\underline{\text{Figures 5(a), 15, 23}}$.
> - AirCraft (High-speed 3D aircraft): $\underline{\text{Figures 5(b), 16}}$.
>
> GeoPT achieves 20% data saving in NASA-CRM and 40% in AirCraft, verifying its generalization.

---

> > ### Author Rebuttal · Reviewer_6Wws · 2026-04-03
> >
> > I thank the authors for providing clarifications to my concerns. I find the rebuttal satisfactory. I raise my rating,

---

> > > ### Author Response · Authors · 2026-04-03
> > >
> > > Dear Reviewer 6Wws,
> > >
> > > We sincerely appreciate your positive feedback on our rebuttal and raising the score. We will include all the rebuttals in the revised paper. Thanks again for taking the time to share your thoughtful review.

---

### Decision · Program_Chairs · 2026-04-30

**Decision:**

Accept (regular)

**Comment:**

This paper lifted geometric pre-training with synthetic dynamics and improved data-efficient neural simulation across industrial fluid and solid benchmarks.
Reviewers agree the paper addresses an important simulation-data bottleneck with a novel, well-presented idea and strong empirical gains.
Initial concerns centered on limited theoretical justification, missing uncertainty estimates, heuristic task-specific velocity, limited evidence for architecture-agnostic and transient claims, and possible underfitting on sensitive aerodynamic cases.
Since the rebuttal resolved the major concerns (added theory, statistics, and cross-backbone evidence), I would suggest the acceptance (scores 4/5/5).